# The Complexity of Correlated Equilibria in Generalized Games

**Martino Bernasconi**
Bocconi University
`martino.bernasconi@unibocconi.it`

**Matteo Castiglioni**
Politecnico di Milano
`matteo.castiglioni@polimi.it`

**Andrea Celli**
Bocconi University
`andrea.celli2@unibocconi.it`

**Gabriele Farina**
Massachusetts Institute of Technology
`gfarina@mit.edu`

## Abstract

Correlated equilibria—and their generalizations known as $\Phi$-equilibria—are a fundamental object of study in game theory, offering a more tractable alternative to Nash equilibria in multi-player settings. While computational aspects of equilibrium computation are well-understood in some settings, fundamental questions are still open in *generalized games*, that is, games in which the set of strategies allowed to each player depends on the other players' strategies. These classes of games model fundamental settings in economics, and have been a cornerstone of economics research since the seminal paper of Arrow and Debreu [1954]. Recently, there has been growing interest, both in economics and in computer science, in studying correlated equilibria in generalized games. It is known that finding a social welfare maximizing correlated equilibrium in generalized games is NP-hard. However, the existence of efficient algorithms to find *any* equilibrium remains an important open question. In this paper, we answer this question in the negative, showing that this problem is PPAD-complete.

## 1 Introduction

Game Theory is a fundamental tool to study the interaction between rational agents. From an application point of view, it has found important applications such as training GANs [Goodfellow et al., 2020], auctions [Nisan et al., 2007], and superhuman performance in strategic games [Brown and Sandholm, 2019, FAIR et al., 2022]. In a series of papers, originating from Papadimitriou [1994] and culminating in Daskalakis et al. [2009], Chen et al. [2009], the problem of computing a Nash equilibrium in multiplayer games was proven to be complete for the complexity class PPAD, which contains problems for which no efficient algorithm is known. On the other hand, correlated equilibria [Aumann, 1987], coarse-correlated equilibria [Moulin and Vial, 1978], and their generalization known as $\Phi$-equilibria [Greenwald and Jafari, 2003], are a relaxation of Nash equilibria, and provide a computationally tractable alternative to Nash in most instances.

The seminal result of Nash [1951] for the existence of equilibrium in non-cooperative games was extended to *generalized games* (also known as pseudo-games or abstract economies) by Arrow and Debreu [1954] and Rosen [1965]. In these games, the set of feasible strategies of one player depends on the strategies played by the other players.

Coupled constraints between players model various applications involving finite resources, where consumption depends on the joint actions of the players. For example, generalized games have been applied to dynamic pricing [Adida and Perakis, 2010], electricity markets [Hobbs and Pang, 2007], and Fisher markets [Brânzei et al., 2014, Conitzer et al., 2022]. A particularly relevant example of

39th Conference on Neural Information Processing Systems (NeurIPS 2025).

joint constraints is bidding in repeated auctions under budget constraints, where budget consumption depends on the joint bids of all players. It is well known that when all players follow no-regret strategies, their average play converges to a correlated equilibrium [Cesa-Bianchi and Lugosi, 2006]. In online advertising, where many companies deploy automated bidders that act as "cost-aware" regret minimizers on behalf of advertisers [Balseiro and Gur, 2019, Aggarwal et al., 2019], it becomes particularly important to ask whether similar convergence guarantees hold for "cost-aware" no-regret algorithms (such as the algorithms presented in Mannor et al. [2009], Mahdavi et al. [2012]).

Thus, an important and natural question is to determine if there are efficient algorithms that compute correlated equilibria in generalized games. While Bernasconi et al. [2023] proved the computational infeasibility of computing a social welfare maximizing correlated equilibrium in generalized games, the complexity of computing *any* equilibrium remains open (see also Zhang et al. [2025]). In this paper, we resolve this open question.

## 1.1 Contributions and Overview of Techniques

The main contribution of this paper is to show that the problem of computing an approximate $\Phi$-equilibrium in generalized games (as formalized in the CONSTRAINED-$\Phi$-EQUILIBRIUM problem in Section 3) is PPAD-complete. The hardness holds for approximation factors that are inverse-polynomial in the instance both in (i) games with two players, and (ii) games with an arbitrary number of players even for a constant number of actions and constraints per player. By providing this characterization, our paper puts to rest a recent line of inquiry in the literature [Bernasconi et al., 2023, Zhang et al., 2025].

At a fundamental level, the question of the complexity of correlated equilibria in generalized games boils down to the tension between two opposing forces:

1. On the one hand, the introduction of *correlation* —as opposed to the *independence* between the play of different players required by Nash equilibria— is a major driver of computational feasibility. For instance, while Nash equilibria are believed to be computationally intractable, correlated solution concepts [Aumann, 1987] are typically computable in polynomial time, even in games with exponentially many actions [Papadimitriou and Roughgarden, 2005, Farina and Pipis, 2024, Daskalakis et al., 2024].

2. On the other hand, the introduction of coupled constraints often has adverse effects on computation [Daskalakis et al., 2021, Papadimitriou et al., 2023, Bernasconi et al., 2024, Anagnostides et al., 2025], even when the constraints are *simple*, *e.g.*, when they define a polytope in the joint strategy space.

Thus, a natural and fundamental question that tries to resolve this tension is the following:

*Which of the two forces prevails? Does correlation remain beneficial in the face of constraints? Or are constraints so detrimental to push the complexity of correlated equilibria in generalized games to the realm of intractability?*

We resolve this question by showing that correlation is not a sufficient relaxation in the presence of coupled constraints. We prove this result by a reduction from the problem of computing Nash equilibria in polymatrix games. At an intuitive level, we could say that the computation of linearly constrained correlated equilibria is indistinguishable from the computation of unconstrained Nash (*i.e.*, independent) equilibria. In other words, the combination of correlation and constraints can simulate independence without constraints.

At the technical level, our main tool is to introduce a "left" and "right" copy of each player in the polymatrix instance, thus creating two teams of players. Each player on one team then plays only with players on the opposing team in a team zero-sum game. We then exploit the constraints to couple the players of the left team to the players of the right team by asking that their marginals coincide. This allows us to essentially remove the terms that are linear in the correlated strategy and only keep the ones that are linear in the marginals (which now resemble the ones of Nash equilibria).

One additional implication of our result is that the quasi-polynomial time algorithm for computing approximate $\Phi$-equilibria of Bernasconi et al. [2023] is tight. Indeed, under the Exponential Time Hypothesis for PPAD (introduced by Rubinstein [2017]), we show that the problem of computing

a (coarse) correlated equilibria in generalized games requires quasi-polynomial time, even with a constant number of players and a constant violation of the incentive compatibility constraints.

Furthermore, we emphasize that although numerous works have established positive convergence guarantees for regret minimization in the unconstrained setting [*e.g.*, Hart and Mas-Colell, 2000], our results demonstrate that no efficient no-regret algorithms can, in general, converge to constrained correlated equilibria. This impossibility has broad implications for systems of constrained no-regret learners, as it precludes convergence unless specific properties of the problem are exploited. A concrete example that has recently attracted significant theoretical and empirical attention is that of Internet advertising platforms, where automated bidding agents are typically operated by the platform itself (see, *e.g.*, [Agarwal et al., 2014, Balseiro and Gur, 2019, Balseiro et al., 2021, Paes Leme et al., 2024]). In such environments, convergence to an equilibrium is often desirable since it promotes stability, predictability, and alignment with the advertisers' individual incentives. Our result, however, shows that this goal cannot be achieved in a "black-box" manner. To design autobidders with convergence guarantees, one must develop algorithms that exploit the specific structural properties of the problem, since no general-purpose approach can ensure convergence in this setting.

## 2 Preliminaries

### 2.1 Multi-player Games and $\Phi$-equilibrium

We consider an $n$-player game. Each player has a set $A = [\ell]$ of actions, and for each player $i \in [n]$ and each tuple of actions $\boldsymbol{a} \in \mathcal{A} := A^n$, the utility of player $i$ is $u_i(\boldsymbol{a}) \in [0, 1]$.[1] A $\Phi$-equilibrium is defined on correlated strategies $z \in \Delta(\mathcal{A})$. In particular, a mediator samples actions $\boldsymbol{a} = (a_1, \ldots, a_n) \sim z$ and communicates to each player $i$ its own action $a_i$. Conditioned on the action recommendation, each player can now decide how to deviate. In particular, each player can pick a function $\phi \in \Phi$ that prescribes its deviation from a set of functions $\Phi$. More specifically, each function $\phi \in \Phi$ maps each action $a \in A$ into a randomized action $\phi(a) \in \Delta(A)$. Note that each function $\phi$ can be represented by a $\ell \times \ell$-dimensional right stochastic matrix on which each row corresponds to a specific action, *i.e.* $\phi(a, b)$ is the probability of playing $b$ when the player is recommended action $a$. By applying a deviation $\phi$ to a correlated strategy $z$ the $i$-th player induces a distribution $\phi \circ_i z$ on $\mathcal{A}$. Formally, for all $\boldsymbol{a} = (a_i, \boldsymbol{a}_{-i}) \in \mathcal{A}$ we let

$$(\phi \circ_i z)(\boldsymbol{a}) = \sum_{b \in A} \phi(b, a_i) z(b, \boldsymbol{a}_{-i}).$$

Moreover, for a function $F : \mathcal{A} \to \mathbb{R}$ and a distribution $z \in \Delta(\mathcal{A})$, with abuse of notation we write $F(z)$ to denote the expected value of $F$ under the distribution $z$, *i.e.* $\sum_{\boldsymbol{a} \in \mathcal{A}} F(\boldsymbol{a}) z(\boldsymbol{a})$. For instance, we denote the expected utility of player $i$ under distribution $z$ as $u_i(z) := \sum_{a \in \mathcal{A}} z(\boldsymbol{a}) u_i(\boldsymbol{a})$.

An $\epsilon$-$\Phi$-equilibrium is a correlated distribution over strategies such that

$$u_i(z) \geq u_i(\phi \circ_i z) - \epsilon \quad \forall i \in [n], \phi \in \Phi.$$

We call a $\Phi$-equilibrium a $0$-$\Phi$-equilibrium. In Appendix A we formally write the sets of functions $\Phi_{\text{CE}}$ and $\Phi_{\text{CCE}}$, which define correlated (CE) and coarse-correlated equilibria (CCE), respectively. Intuitively, CCEs deviations are deviations that cannot depend on the recommended action, while CE deviations are a larger set where deviations could be different based on the recommended action.

### 2.2 Constrained $\Phi$-equilibria

To model situations in which there are common shared resources, we introduce costs that depend on the joint actions of all players. In particular, each player $i \in [n]$ has $m$ costs $\{C_i^j(\boldsymbol{a})\}_{j \in [m]} \in [-1, 1]^m$ associated with each tuple of action $\boldsymbol{a} \in \mathcal{A}$. For any $\nu \geq 0$, we define as $\mathcal{S}_i^\nu$ the set of correlated strategies that are $\nu$-safe (in expectation) for the $i$-th player, *i.e.*,

$$\mathcal{S}_i^\nu := \left\{ z \in \Delta(\mathcal{A}) : C_i^j(z) \leq \nu \quad \forall j \in [m] \right\}.$$

Intuitively, $\mathcal{S}_i^\nu$ guarantees that the expected cost of player $i$ is at most $\nu$ for each of its resources. Moreover, we define the set $\mathcal{S}^\nu = \cap_{i \in [n]} \mathcal{S}_i^\nu$ as the set of strategies that are $\nu$-safe for all players. For

---

[1]Given an integer $i \in \mathbb{N}$, we denote with $[i]$ the set $\{1, \ldots, i\}$.

each $z \in \mathcal{S}^\nu$ and player $i \in [n]$ we can define the set of *safe*-deviations, *i.e.*,

$$\Phi_i^{\mathcal{S}}(z) := \{\phi \in \Phi : (\phi \circ_i z) \in \mathcal{S}_i\}.$$

To guarantee that the game is well-defined and the existence of solutions, we make the following mild assumption. This assumption requires that all agents always have a deviation that ensures that cost constraints are satisfied. Formally, for all $z \in \Delta(\mathcal{A})$ and $i \in [n]$, it holds that $\Phi_i^{\mathcal{S}}(z) \neq \emptyset$. This assumption is required to guarantee that players always have safe strategies, and it is common in the literature. Bernasconi et al. [2023] proved existence (but not PPAD-membership) with a stronger notion related to strict feasibility, which was then removed in subsequent works [Boufous et al., 2024, Ni et al., 2025]. This assumption is generally satisfied, as it is reasonable to let players have a void action with zero utility and zero cost, irrespective of other players' strategies. We say that a correlated strategy $z \in \mathcal{S}^\nu$ is a Constrained $(\epsilon, \nu)$-$\Phi$-equilibrium if it holds that $u_i(z) \geq u_i(\phi \circ_i z) - \epsilon$ for all players $i \in [n]$ and deviations in $\phi \in \Phi_i^{\mathcal{S}}(z)$. Intuitively, ignoring the technicalities related to the relaxation $\nu$, a correlated strategy $z$ is a constrained $\epsilon$-$\Phi$-equilibrium if $z$ is safe for each player and each player cannot earn more than $\epsilon$ by applying a safe deviation to the correlated strategy $z$.

## 2.3 Polymatrix and team games

Polymatrix games are a type of multiplayer games in which the $n$ players interact only with a subset of other players and the game between any two players is a bi-matrix game. Formally, we can introduce the following computational problem:

> **POLYMATRIX**
> **Input:** a graph $G = (V, E)$, a matrix $A^{i,j} \in [0,1]^{k \times k}$ for each $(i, j) \in E$, and an approximation $\epsilon > 0$.
>
> **Output:** vectors $x_i \in \Delta([k])$, $i \in V$ such that:
> $$\sum_{j \in V:(i,j) \in E} x_i^\top A^{i,j} x_j \geq \sum_{j \in V:(i,j) \in E} \tilde{x}_i^\top A^{i,j} x_j - \epsilon \quad \forall i \in V, \tilde{x}_i \in \Delta([k]).$$

In a breakthrough result Rubinstein [2015] (later improved by Deligkas et al. [2022]) proved constant inapproximability of POLYMATRIX for constant number of actions and degree of the graph.

**Theorem 2.1** (Rubinstein [2015, Theorem 1])**.** *There exists a constant $\epsilon^*$ such that* POLYMATRIX *is* PPAD-*complete, even when the graph has degree* 3 *and each player has* 2 *actions.*

An important subclass of polymatrix games are *two-team zero-sum game*s, in which $V$ can be partitioned into two teams such that: (i) each edge between players of different teams is a zero-sum game, and (ii) each edge between players of the same team is a coordination game. If one of the teams has no internal coordination edges, these games are called *two-team polymatrix zero-sum game with independent adversaries* [Hollender et al., 2025]. Here we also consider the case in which there are no coordination games in between both teams, *i.e.*, the graph $G$ is fully bipartite. We call these *two-team polymatrix zero-sum game with independent teams*. These games are special cases of zero-sum polymatrix games that admit LP-based polynomial-time algorithms [Cai et al., 2016].

## 2.4 PPAD-hardness and Total Problems

The class PPAD is an important subclass of total search problems *i.e.*, TFNP, which includes search problems with solutions that can be checked in polynomial time. PPAD was introduced by Papadimitriou [1994] to capture problems whose totality is guaranteed by parity-like arguments and, equivalently, by fixed-point arguments such as Brouwer's. PPAD-complete problems are unlikely to be solvable in polynomial time. This is also supported by several lower bounds based on cryptographic assumptions [Bitansky et al., 2015, Garg et al., 2016, Choudhuri et al., 2019].

Moreover, for ease of presentation, we use a more manageable class of total problems, as we allow for promise problems, *i.e.*, problems whose totality is guaranteed only if the instance satisfies some additional properties (promises) whose validity is not clear how to check in polynomial time.[2]

---

[2]Formally, to guarantee the totality of our problems we should accept as solutions a violation of the promise (see, Fearnley et al. [2020, 2022], Hollender [2021] for an in-depth treatment of promise-preserving reductions.

# 3 Reduction from POLYMATRIX to CONSTRAINED-$\Phi$-EQUILIBRIUM

We define a computational search problem associated with constrained $\Phi$-equilibria.

> **CONSTRAINED-$\Phi$-EQUILIBRIUM**
>
> **Input:** $\epsilon' > 0$, $\nu > 0$, an utility function $u_i : \mathcal{A} \to [0,1]$ and $m$ cost functions $C_i^j : \mathcal{A} \to [-1,1]$, $j \in [m]$ for each player $i$, and a set of possible deviations $\Phi$ (given as a list of linear inequalities, defining a polytope of right stochastic matrices) with $A \in \mathbb{R}^{k \times 2\ell}$ and $b \in \mathbb{R}^k$.
>
> **Output:** a vector $z \in \mathcal{S}^\nu$ such that $u_i(z) \geq u_i(\phi \circ_i z) - \epsilon' \quad \forall i \in [n], \phi \in \Phi_i^{\mathcal{S}}(z)$.
>
> **Promise:** It is promised that $\Phi_i^{\mathcal{S}}(z) \neq \emptyset$ for all $z \in \Delta(\mathcal{A})$ and $i \in [n]$.

Formally, to fix a representation of the output, we think of $z$ as a mixture of product distributions. Notice that such a representation is essential to guarantee a representation polynomial in the support and independent of $|\mathcal{A}|$.

In the following, we will provide a reduction from POLYMATRIX to CONSTRAINED-$\Phi$-EQUILIBRIUM.

**Intuitive high-level idea of the proof** To give a high-level intuition of the reduction, we sketch a simplified reduction from the problem of computing Nash equilibria in two-player general-sum games. Given a general-sum game $A, B \in [0,1]^{k \times k}$, we can write a feasibility LP for coarse correlated equilibria as follows:[3]

$$\text{Find } z \text{ such that:} \begin{cases} A(z) \geq A(\tilde{x} \otimes z_{-1}) \, \forall \tilde{x} \in \Delta([k]) & \text{(1a)} \\ B(z) \geq B(z_{-2} \otimes \tilde{y}) \, \forall \tilde{y} \in \Delta([k]) & \text{(1b)} \\ z \in \Delta([k]^2) \end{cases}$$

If we were to add a constraint forcing $z$ to be a product distribution, *i.e.*, $z = x \otimes y$, then Equation (1) would turn into a feasibility problem whose solutions are Nash equilibria of $A, B$.[4] This (non-linear) constraint is exactly what makes computing Nash equilibria hard. Thus, ideally, we would like to impose constraints to force the correlated strategy to be a product distribution. However, the constraints we can employ are linear in $z$, and we cannot follow this idea directly.

By observing Problem 1, we can see that the right-hand sides of the two incentive compatibility (IC) constraints (1a) and (1b), are linear only in the marginals. Thus, it could be convenient to: $(i)$ impose the costs to the $x$ player such that the marginal of the $x$ player copies the marginal of the $y$ player, *i.e.*, $z_{-1} = z_{-2}$ (note that this is a linear constraint in $z$), $(ii)$ take $A = -B$ so to erase the terms linear in $z$ when summing the two IC constraints of Equation (1).

It is easy to see that, due to $(i)$, we also have that the deviations $\tilde{x}$ admissible to the $x$ player are just $\{z_{-1}\}$. On the other hand, there are no costs on the second player, and thus the deviations $\tilde{y}$ are free. Define $h = z_{-1} = z_{-2}$. Then summing (1a) and (1b) we obtain

$$A(z) + B(z) \geq A(h \otimes h) + B(h \otimes \tilde{y}) \stackrel{\text{Due to } (ii)}{\Longrightarrow} h^\top B h \geq h^\top B \tilde{y} \quad \forall \tilde{y} \in \Delta([k]).$$

It is easy to choose $B$ such that it is PPAD-hard to find a solution to the right-hand side inequality above. For example, we can choose $B = \begin{bmatrix} 0 & \tilde{A} \\ \tilde{B}^\top & 0 \end{bmatrix}$ for another two-player general-sum game $\tilde{A}, \tilde{B}$. This choice introduces two additional players for each player of the original game, essentially creating two opposing teams playing a zero-sum game.

The above construction is informal and skips many of the technical challenges, such as the fact that we are only allowed to use constraints that are satisfied approximately. Also, we derive a more general result and broader consequences by reducing from POLYMATRIX instead of computing Nash in two-player games (see Section 3.2). This is indeed one central part of the formal reduction that we will present in the next section.

Given a graph $G = (V, E)$, we denote with $\deg(G)$ the maximum degree of graph $G$. Then, the main result of this section reads as follows:

---

[3]We denote with $z_{-i}$ the marginalization of $z$ with respect to the $i$-th player.
[4]Note that $A(x \otimes y) = x^\top A y$.

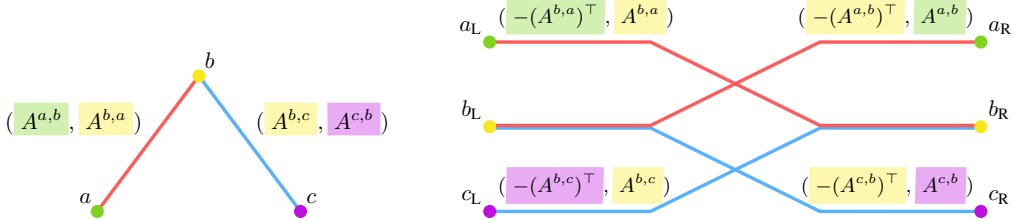

(a) Example of a POLYMATRIX instance with three players $V = \{a, b, c\}$.

(b) Corresponding instance of CONSTRAINED-$\Phi$-EQUILIBRIUM with $\mathcal{N}_\mathrm{L} = \{a_\mathrm{L}, b_\mathrm{L}, c_\mathrm{L}\}$ and $\mathcal{N}_\mathrm{R} = \{a_\mathrm{R}, b_\mathrm{R}, c_\mathrm{R}\}$.

Figure 1: On the edges, we reported the matrix corresponding to the utilities of the players. Colors indicate the row player of the bi-matrix game and the player associated with the utility that is associated with the matrix.

**Theorem 3.1.** *There is a polynomial-time reduction from instances $G = (V, E)$ of POLYMA-TRIX with $n$ players, $k$ actions per player, and approximation $\epsilon$ to instances of CONSTRAINED-$\Phi$-EQUILIBRIUM with approximation $\epsilon' = O(\frac{\epsilon}{n})$ and $\nu = O(\frac{\epsilon}{nk \deg(G)})$ and CCE deviations.*

### 3.1 Proof of Theorem 3.1

**Construction** Given an instance of POLYMATRIX with graph $G = (V, E)$, matrices $\{A^{i,j}\}_{(i,j)\in E}$ and approximation $\epsilon$, we build an instance of CONSTRAINED-$\Phi$-EQUILIBRIUM as follows:

▶ Set of players $\mathcal{N} = \mathcal{N}_\mathrm{L} \cup \mathcal{N}_\mathrm{R}$, where $\mathcal{N}_\mathrm{L} = \{i_\mathrm{L} : i \in V\}$ and $\mathcal{N}_\mathrm{R} = \{i_\mathrm{R} : i \in V\}$;

▶ For each player in $\mathcal{N}$ the set of actions is $A = [k]$ as in the original POLYMATRIX instance;

▶ For every $i \in V$, the utility of a player $i_\mathrm{L} \in \mathcal{N}_\mathrm{L}$, under an action profile $\boldsymbol{a} \in A^{|\mathcal{N}|}$, is defined as $u_{i_\mathrm{L}}(\boldsymbol{a}) = -\sum_{j:(i,j)\in E}(A^{j,i})^\top(a_{i_\mathrm{L}}, a_{j_\mathrm{R}})$, while the utility of a player $i_\mathrm{R} \in \mathcal{N}_\mathrm{R}$ under action profile $\boldsymbol{a}$ is $u_{i_\mathrm{R}}(\boldsymbol{a}) = \sum_{j:(i,j)\in E} A^{i,j}(a_{i_\mathrm{R}}, a_{j_\mathrm{L}})$;

▶ The set of deviations is $\Phi_\mathrm{CCE}$ (see Appendix A for a definition);

▶ For each player $i_\mathrm{L}$ there are $2k$ costs. For each $j \in [2k]$, let

$$C_{i_\mathrm{L}}^j(\boldsymbol{a}) = \begin{cases} 2\mathbb{I}(j \le k) - 1 & \text{if } a_{i_\mathrm{L}} = j \mod k \text{ and } a_{i_\mathrm{R}} \ne j \mod k \\ 2\mathbb{I}(j > k) - 1 & \text{if } a_{i_\mathrm{R}} \ne j \mod k \text{ and } a_{i_\mathrm{L}} = j \mod k \;; \\ 0 & \text{otherwise} \end{cases}$$

For more intuition on the construction, we refer to Figure 1.

**Properties of the instance** The instance of CONSTRAINED-$\Phi$-EQUILIBRIUM resulting from this construction has twice as many players as the original instance, and the same degree number of actions per player $k$ as the original instance. Moreover, the CONSTRAINED-$\Phi$-EQUILIBRIUM instance guarantees that each player has at most $2k$ costs.

The game resulting from the reduction is a two-team polymatrix zero-sum game with independent teams. Indeed, we implicitly defined a graph $\widetilde{G} = (\mathcal{N}, \widetilde{E})$ which has edges only between players $\mathcal{N}_\mathrm{L}$ and $\mathcal{N}_\mathrm{R}$. More specifically, an edge is included between $i_\mathrm{L} \in \mathcal{N}_\mathrm{L}$ and $j_\mathrm{R} \in \mathcal{N}_\mathrm{R}$ if and only if there was an edge between $i$ and $j$ in the original POLYMATRIX instance. Moreover, the game is zero-sum. Indeed, for each couple of players $i_\mathrm{L} \in \mathcal{N}_\mathrm{L}$ and $j_\mathrm{R} \in \mathcal{N}_\mathrm{R}$, the utilities are given by payoff matrices $\tilde{A}^{i_\mathrm{L}, j_\mathrm{R}} = -(A^{j,i})^\top$ and $\tilde{A}^{i_\mathrm{R}, j_\mathrm{L}} = A^{i,j}$, respectively. Finally, the costs of player $i_\mathrm{L}$ only depend on the actions of player $i_\mathrm{R}$.

Summing over the utilities of all players on each side, we get the following lemma:

**Lemma 3.2.** *For each tuple $\boldsymbol{a} \in \mathcal{A}$, define the utility of the "left team" as $u_\mathrm{L}(\boldsymbol{a}) = \sum_{i\in V} u_{i_\mathrm{L}}(\boldsymbol{a})$ and the utility of the "right team" as $u_\mathrm{R}(a) = \sum_{i\in V} u_{i_\mathrm{R}}(a)$. Then $u_\mathrm{L}(\boldsymbol{a}) = -u_\mathrm{R}(\boldsymbol{a})$.*

**Marginalization** Before delving into the correctness of the reduction, we need to introduce some additional notation that helps bridge between correlated equilibria and Nash equilibria. For any

subset of players $\widetilde{\mathcal{N}} \subseteq \mathcal{N}$ and any tuple of actions $\boldsymbol{a} \in \mathcal{A}$, we denote with $\boldsymbol{a}_{\widetilde{\mathcal{N}}}$ the actions pertaining to players in $\widetilde{\mathcal{N}}$. We also denote with $-\widetilde{\mathcal{N}} = \mathcal{N} \setminus \widetilde{\mathcal{N}}$ the complement of $\widetilde{\mathcal{N}}$.

Now, we can define the marginalization of the correlated strategy $z$ restricted to a set of players $\widetilde{\mathcal{N}}$ by summing over the actions of all other players. Formally, we write

$$m_{\widetilde{\mathcal{N}}}(z|\boldsymbol{a}_{\widetilde{\mathcal{N}}}) = \sum_{\boldsymbol{a}_{-\widetilde{\mathcal{N}}} \in A^{|\mathcal{N} \setminus \widetilde{\mathcal{N}}|}} z(\boldsymbol{a}_{\widetilde{\mathcal{N}}}, \boldsymbol{a}_{-\widetilde{\mathcal{N}}}).$$

We can then consider $m_{\widetilde{\mathcal{N}}}(z)$ as a vector indexed by all the tuples $\boldsymbol{a}_{\widetilde{\mathcal{N}}} \in A^{|\widetilde{\mathcal{N}}|}$ which, clearly, is a distribution over $A^{|\widetilde{\mathcal{N}}|}$.

**Effects on the costs** We now prove the main properties related to costs. Intuitively, every safe strategy guarantees that the marginals of players $i_{\mathrm{L}}$ are close to the marginals of the players $i_{\mathrm{R}}$ for each $i \in V$. In particular, each action $j$ has two associated costs $C_{i_{\mathrm{L}}}^{j}$ and $C_{i_{\mathrm{L}}}^{j+k}$ that force the marginal of the first player to mimic the marginal of the second player. Namely,

**Lemma 3.3.** *Given a $z \in \mathcal{S}^{\nu}$, and $i \in V$, it holds that $\|m_{i_{\mathrm{L}}}(z) - m_{i_{\mathrm{R}}}(z)\|_{\infty} \leq \nu$. Moreover, given a $z \in \Delta(A^{|\mathcal{N}|})$ and an $i \in V$, the set of safe deviations of player $i_{\mathrm{L}}$ is $\Phi_{i_{\mathrm{L}}}^{\mathcal{S}}(z) = \{x_{i_{\mathrm{L}}} : \|x_{i_{\mathrm{L}}} - m_{i_{\mathrm{R}}}(z)\|_{\infty} \leq 0\}$, which guarantees $\Phi_{i_{\mathrm{L}}}^{\mathcal{S}}(z) \neq \emptyset$.*

**Correctness** We recall that $k = |A|$ is the number of actions of each player and $n = |V|$ is the number of players. Take any $z \in \mathcal{S}^{\nu}$ that is a solution to CONSTRAINED-$\Phi$-EQUILIBRIUM, combining it with Lemma 3.3, we get that for all $i \in V$ it holds:

$$u_{i_{\mathrm{L}}}(z) \geq u_{i_{\mathrm{L}}}(x_{i_{\mathrm{L}}} \otimes m_{\mathcal{N} \setminus i_{\mathrm{L}}}(z)) - \epsilon' \quad \forall x_{i_{\mathrm{L}}} \in \Delta(A) : \|x_{i_{\mathrm{L}}} - m_{i_{\mathrm{R}}}(z)\|_{\infty} \leq 0 \tag{2}$$

and

$$u_{i_{\mathrm{R}}}(z) \geq u_{i_{\mathrm{R}}}(x_{i_{\mathrm{R}}} \otimes m_{\mathcal{N} \setminus i_{\mathrm{R}}}(z)) - \epsilon' \quad \forall x_{i_{\mathrm{R}}} \in \Delta(A). \tag{3}$$

Notice that we can write explicitly the right-hand side of Equation (2) as

$$u_{i_{\mathrm{L}}}(x_{i_{\mathrm{L}}} \otimes m_{\mathcal{N} \setminus i_{\mathrm{L}}}(z)) = - \sum_{j : (i,j) \in E} x_{i_{\mathrm{L}}}^{\top} (A^{j,i})^{\top} m_{j_{\mathrm{R}}}(z) = - \sum_{j : (i,j) \in E} m_{j_{\mathrm{R}}}(z)^{\top} A^{j,i} x_{i_{\mathrm{L}}},$$

We then specialize Equation (2) for $x_{i_{\mathrm{L}}} = m_{i_{\mathrm{R}}}(z)$ and sum over all $i \in V$. This results in the following inequality (where we also swapped the identity of $i$ and $j$ in the last equality):

$$u_{\mathrm{L}}(z) \geq - \sum_{(i,j) \in E} m_{j_{\mathrm{R}}}(z)^{\top} A^{j,i} m_{i_{\mathrm{R}}}(z) - n\epsilon' = - \sum_{(i,j) \in E} m_{i_{\mathrm{R}}}(z)^{\top} A^{i,j} m_{j_{\mathrm{R}}}(z) - n\epsilon'. \tag{4}$$

Similarly, we analyze Equation (3) and use the first statement of Lemma 3.3:

$$u_{\mathrm{R}}(z) \geq \sum_{i \in V} u_{i_{\mathrm{R}}}(x_{i_{\mathrm{R}}} \otimes m_{\mathcal{N} \setminus i_{\mathrm{R}}}(z)) - n\epsilon'$$

$$= \sum_{(i,j) \in E} x_{i_{\mathrm{R}}}^{\top} A^{i,j} m_{j_{\mathrm{L}}}(z) - n\epsilon'$$

$$= \sum_{(i,j) \in E} x_{i_{\mathrm{R}}}^{\top} A^{i,j} m_{j_{\mathrm{R}}}(z) + \sum_{i \in V} \sum_{j : (i,j) \in E} x_{i_{\mathrm{R}}}^{\top} A^{i,j} (m_{j_{\mathrm{L}}}(z) - m_{j_{\mathrm{R}}}(z)) - n\epsilon'$$

$$\geq \sum_{(i,j) \in E} x_{i_{\mathrm{R}}}^{\top} A^{i,j} m_{j_{\mathrm{R}}}(z) - n(\nu k \deg(G) + \epsilon') \quad \forall \{x_{i_{\mathrm{R}}}\}_{i \in V} \in \Delta(A)^{n}. \tag{5}$$

Summing Equation (4) and Equation (5), and using the fact that $u_{\mathrm{R}}(z) = -u_{\mathrm{L}}(z)$ from Lemma 3.2, we obtain that

$$\sum_{(i,j) \in E} m_{i_{\mathrm{R}}}(z)^{\top} A^{i,j} m_{j_{\mathrm{R}}}(z) \geq \sum_{(i,j) \in E} x_{i_{\mathrm{R}}}^{\top} A^{i,j} m_{j_{\mathrm{R}}}(z) - n(\nu k \deg(G) + 2\epsilon') \quad \forall \{x_{i_{\mathrm{R}}}\}_{i \in V} \in \Delta(A)^{n}.$$

Which, for all players $i \in V$ can be specialized to

$$\sum_{j : (i,j) \in E} m_{i_{\mathrm{R}}}(z)^{\top} A^{i,j} m_{j_{\mathrm{R}}}(z) \geq \sum_{j : (i,j) \in E} x_{i_{\mathrm{R}}}^{\top} A^{i,j} m_{j_{\mathrm{R}}}(z) - n(\nu k \deg(G) + 2\epsilon') \quad \forall x_{i_{\mathrm{R}}} \in \Delta(A),$$

by setting $x_{j_{\mathrm{R}}} = m_{j_{\mathrm{R}}}(z)$ for all $j \neq i$.

Thus, taking $h_i = m_{i_{\mathrm{R}}}(z)$ for all players $i \in V$, and $\epsilon' = \epsilon/4n$ and $\nu = \epsilon/(2nk \deg(G))$ we prove that $h_i$ is a solution of POLYMATRIX with approximation $\epsilon$.

## 3.2 Implications of the PPAD-hardness

The class of deviations used in the reduction of Theorem 3.1 is $\Phi_{\text{CCE}}$, which is a subset of the deviations of $\Phi_{\text{CE}}$, thus, our hardness works also for correlated equilibrium in generalized games.

We now discuss some special cases in which our problem is PPAD-hard. POLYMATRIX is PPAD-hard even when $\epsilon, \deg(G), k = O(1)$ (see Theorem 2.1). Thus, Theorem 3.1 implies that CONSTRAINED-$\Phi$-EQUILIBRIUM is hard when each player has $m = O(1)$ constraints and actions, when the slackness $\nu$ and approximation $\epsilon$ are $O(n^{-1})$.

Moreover, the instances resulting from our reduction form a bipartite graph and, more specifically, a two-team zero-sum polymatrix game with independent teams, as defined in Section 2.3. Importantly, for this case of two-team zero-sum polymatrix game with no constraints, there are known LP-based polynomial-time algorithms [Cai et al., 2016]. It is an interesting open question whether it is possible to prove hardness for constant approximations, arbitrary number of players, constant number of actions per player, and constant number of constraints per player.

We can also consider the bi-matrix instance of POLYMATRIX in which we have only 2 players and an arbitrary number of actions $k$. This shows that our problem is PPAD-hard even with a constant number of players when $\epsilon' = O(1)$ and $\nu = O(k^{-1})$.

Moreover, assuming the Exponential Time Hypothesis for PPAD (namely that solving the canonical problem END-OF-THE-LINE requires exponential time), Rubinstein [2017] showed that it must take at least $k^{\log^{1-o(1)}(k)}$ time to find a constant approximation for any $k \times k$ bi-matrix game. This imposes a quasi-polynomial lower bound on the CONSTRAINED-$\Phi$-EQUILIBRIUM problem for constant $\epsilon$ and $\nu = \text{poly}(k^{-1})$. On the other hand, Bernasconi et al. [2023, Corollary 4.3] provides a quasi-polynomial time algorithm for constant approximation and a constant number of players in $k^{\log(k)}$ time, thus essentially closing the gap.

## 4 PPAD-membership

We now introduce a computational problem related to quasi-variational inequalities that will serve as the basis for our membership reduction.

> **QUASIVI**
>
> **Inputs:** $G, L, \epsilon > 0, \nu > 0$, a circuit implementing a $G$-Lipschitz function $F : \mathbb{R}^d \to \mathbb{R}^d$, and two circuits implementing a $L$-Lipschitz continuous matrix valued function $A : [0,1]^d \to \mathbb{R}^{n \times d}$ and a $L$-Lipschitz continuous vector valued function $b : \mathbb{R}^n \to \mathbb{R}^d$ defining the correspondence $Q_\nu(\tilde{z}) := \{z \in [0,1]^d : A(\tilde{z})z \le b(\tilde{z}) + \nu 1_d\}$.[a]
>
> **Output:** a point $z \in Q_\nu(z)$ such that $F(z)^\top(\tilde{z} - z) \ge -\epsilon$, for all $\tilde{z} \in Q_\nu(z)$.
>
> **Promise:** The correspondence is promised to satisfy $Q_0(z) \ne \emptyset$ for all $z \in [0,1]^d$, the function $F$ is $G$-Lipschitz and the correspondence $Q_0(z)$ is $L$-Lipschitz.
>
> ---
> [a]We say that a matrix-valued function $A : [0,1]^d \to \mathbb{R}^{n \times d}$ is $L$-Lipschitz if the function $\tilde{z} \mapsto A(\tilde{z})z$ is $L$-Lipschitz for every $z \in [0,1]^d$ in the $\ell_2$-norm.

**Theorem 4.1** (Bernasconi et al. [2024, Theorem 3.4]). QUASIVI $\in$ PPAD.

To prove that CONSTRAINED-$\Phi$-EQUILIBRIUM is total and more specifically in PPAD we reduce it to QUASIVI. Our idea is to restrict to Nash-like equilibria with CCE deviations, *i.e.*, restrict to product strategies $\bigtimes_{i \in [n]} p_i \in \Delta(A)^n$, where $p_i \in \Delta(A)$. We will show that these are a subset of constrained $\Phi$-equilibria for any set of deviations $\Phi$.

**Proposition 4.2.** CONSTRAINED-$\Phi$-EQUILIBRIUM $\in$ PPAD.

*Proof.* The proof hinges on showing that for any instance of CONSTRAINED-$\Phi$-EQUILIBRIUM with CCE deviations: (i) we can build an instance of QUASIVI in polynomial time and (ii) from the solution of QUASIVI, we can build a solution to the original CONSTRAINED-$\Phi$-EQUILIBRIUM instance which is a product distribution. We show that considering CCE deviations is enough, since these deviations are the most expressive when applied to product distributions. Indeed, the image

of any product distribution under the set of deviations $\Phi_{\mathrm{CCE}}$ is a superset of the image under any possible set of phi deviations. Formally, given a player $i \in [n]$, a product distribution $p = \bigotimes_{i \in [n]} p_i$, a set of deviations $\Phi$, and a deviation $\phi \in \Phi$, there exists a marginal $\tilde{p}_i$ of the $i$-th player such that $\phi \circ_i p = \tilde{p}_i \otimes p_{-i}$. In particular, we can take $\tilde{p}_i \in \Delta(A)$ such that $\tilde{p}_i(a_i) = \sum_{b \in A} \phi(b, a_i) p_i(b)$ for each $a_i \in A$. Then, a CCE deviation can replicate the marginal $\tilde{p}_i$ by taking $\phi \in \Phi_{\mathrm{CCE}}$ such that each row of the stochastic matrix is $\tilde{p}_i$. This also proves that there exists a safe CCE deviations for every product distribution, from the existence of a safe deviation of the original set of deviations $\Phi$.

Indeed, any instance of CONSTRAINED-$\Phi$-EQUILIBRIUM is guaranteed by assumption to have the set of feasible deviations non-empty for each correlated distribution. As a direct consequence of our previous observation, we have that the set of feasible deviations in CCE is not empty when considering only product distributions $z$. This can be proven by observing that the strategies obtained by applying deviations from any set $\Phi$ to product distributions form a subset of the strategies obtained by applying CCE deviations.

**Construction** To define an instance of QUASIVI, we define a correspondence $Q : \mathbb{R}^d \rightrightarrows \mathbb{R}^d$ and an operator $F : \mathbb{R}^d \to \mathbb{R}^d$ satisfying the appropriate conditions of the QUASIVI problem. Namely, we need the correspondence $Q(\tilde{z})$ to be linear for every $\tilde{z}$ and with Lipschitz coefficients. Moreover, we also require that the operator $F$ is Lipschitz.[5]

Since we only consider product distributions, $\ell n$ numbers suffice to uniquely specify the distribution. Thus, we can "flatten" the product strategies $p = \bigotimes_{i \in [n]} p_i$ into $z = [p_1^\top | \cdots | p_n^\top]^\top$ (equivalently $\tilde{z}$ is the flattening of $\tilde{p}$) and consider the correspondence $Q(\tilde{z}) = \{z \in [0,1]^{\ell n} : A(\tilde{z}) z \le b(\tilde{z})\}$, that encodes both the costs and the simplex constraints. Formally: $A(\tilde{z}) = \mathrm{diag}(A_1(\tilde{z}), \ldots, A_n(\tilde{z})) \in \mathbb{R}^{n(m+2) \times \ell n}$ and $b(\tilde{z}) = [b_1(\tilde{z})^\top | \ldots | b_n(\tilde{z})^\top]^\top \in \mathbb{R}^{n(m+2)}$, where $A_i(z)$ is a block matrix $A_i(\tilde{z}) = \begin{bmatrix} D_i(\tilde{z}) \\ 1_\ell^\top \\ -1_\ell^\top \end{bmatrix} \in \mathbb{R}^{(m+2) \times \ell}$, $b_i(\tilde{z}) = \begin{bmatrix} 0_m \\ 1 \\ -1 \end{bmatrix} \in \mathbb{R}^{m+2}$, and $D_i(\tilde{z}) \in \mathbb{R}^{m \times \ell}$ is such that

$$[0_{m \times \ell}, \cdots, D_i(\tilde{z}), \cdots, 0_{m \times \ell}] z \le 0_m \iff C_i^j(p_i \otimes \tilde{p}_{-i}) \le 0 \; \forall j \in [m].$$

The explicit definition of $D_i(\tilde{z})$ is cumbersome due to the flattening notation and we defer it to Appendix C.1.

It is easy to check that $z \in Q(z)$ if the corresponding "unflattened" product strategies $p$ satisfy the simplex and cost constraints for each player. The next claim shows that the instances created by our reduction satisfy the promises needed by the QUASIVI problem.

**Claim 4.3.** *The functions $\tilde{z} \mapsto A(\tilde{z}) z$ and $\tilde{z} \mapsto b(\tilde{z})^\top z$ defining the correspondence are L-Lipschitz for every $z \in [0,1]^{\ell n}$ where $L$ has a representation polynomial in the size of the instance.*

Now, we build the operator $F$ of the QVI by stacking the gradients of the utilities. For any product distribution $p = \bigotimes_{i \in [n]} p_i$ and flattening $z$ we define, with slight abuse of notation, $F(z) := (-\nabla_{p_1} u_1(p), \ldots, -\nabla_{p_n} u_n(p))$. The following claims that the operator $F$ satisfies the properties required by the QUASIVI problem. This shows that the constructed instance satisfies all the promises of the QUASIVI problem.

**Claim 4.4.** *The operator $F : [0,1]^{\ell n} \to [0,1]^{\ell n}$ is G-Lipschitz where $G$ has a representation polynomial in the size of the instance.*

**Correctness** For every vector $r \in [0,1]^{\ell n}$, define $r^i = r_{[\ell(i-1), \ldots, \ell i - 1]}$, which effectively divides (and "unflattens") the vector $r$ in $n$ components of length $\ell$ each corresponding to the strategies of the $i$-th player. We will use this notation to reconstruct a solution to CONSTRAINED-$\Phi$-EQUILIBRIUM from a solution $z$ of QUASIVI. Indeed, take a solution $z$ of the QUASIVI problem, instantiated with $\epsilon'$ and $\nu'$ later to be defined. We now claim that the renormalization $p = \otimes_{i \in [n]} p_i$ (where

---

[5]Crucially, the Lipschitz constants are inputs of the QUASIVI problem, and thus, we need to make sure that they have a polynomial representation in the original CONSTRAINED-$\Phi$-EQUILIBRIUM instance.

$p_i = z^i/\|z^i\|_1$) is a solution to CONSTRAINED-$\Phi$-EQUILIBRIUM.[6] The only tricky part is simply handling the fact that all the constraints are only approximately satisfied.

**Claim 4.5.** *Let $\nu' = \min(\frac{\epsilon}{2}, \frac{\nu^2}{2n})$ and $\epsilon' = \frac{\epsilon}{2}(1 - n\nu')$, then $p$ is such that*

$$u_i(p) \geq u_i(\tilde{p}_i \otimes p_{-i}) - \epsilon \quad \forall \tilde{p}_i \in \Phi_i^{\mathcal{S}}(p)$$

*and $p \in \mathcal{S}^\nu$.*

This concludes the proof, as it shows that $p$ (which is a product distribution) is a CONSTRAINED-$\Phi$-EQUILIBRIUM for CCE deviations, with approximation $\epsilon$ and slackness $\nu$ and thus a CONSTRAINED-$\Phi$-EQUILIBRIUM for any set $\Phi$. $\qquad\square$

## Acknowledgments

The work of MB and AC was partially funded by the European Union. Views and opinions expressed are however those of the author(s) only and do not necessarily reflect those of the European Union or the European Research Council Executive Agency. Neither the European Union nor the granting authority can be held responsible for them.

MB and AC are supported by an ERC grant (Project 101165466 — PLA-STEER) and by MUR - PRIN 2022 project 2022R45NBB funded by the Next Generation EU program. MC is partially supported by the FAIR (Future Artificial Intelligence Research) project PE0000013, funded by the NextGenerationEU program within the PNRR-PE-AI scheme (M4C2, investment 1.3, line on Artificial Intelligence) and the EU Horizon project ELIAS (European Lighthouse of AI for Sustainability, No. 101120237). GF is supported in part by the National Science Foundation award CCF-2443068, ONR grant N000142512296, and an AI2050 Early Career Fellowship.

The authors also thank Tingting Ni for helpful discussion and suggestions.

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

# A Examples of $\Phi$-equilibria

We now introduce two notable examples of sets $\Phi$, namely Correlated and Coarse Correlated equilibria.

Correlated equilibria (CEs) are obtained by considering all possible deviation strategies, *i.e.*,

$$\Phi_{\mathrm{CE}} = \left\{ \phi \in [0,1]^{\ell \times \ell} : \sum_{b \in A} \phi(a,b) = 1 \quad \forall a \in A \right\}.$$

This set models a player that can observe its own recommendation and deviate to any other action with some probability.

Another important class of equilibria is Coarse Correlated Equilibria (CCEs) that are defined by the set

$$\Phi_{\mathrm{CCE}} = \left\{ \phi \in [0,1]^{\ell \times \ell} : \phi(a,b) = \phi(a',b) \quad \forall a,a' \in A \wedge \sum_{b \in A} \phi(a,b) = 1 \quad \forall a \in A \right\}.$$

This models a player whose deviations are forced to be equal for each recommended action $a \in A$. Intuitively, the player has to decide their own deviation strategy before seeing the recommended action $a$. This greatly simplifies the possible deviation since $\phi \in \Phi_{\mathrm{CCE}}$ can simply be identified with the marginals it induces.

# B Missing proof from Section 3 (hardness)

**Lemma 3.2.** *For each tuple $\boldsymbol{a} \in \mathcal{A}$, define the utility of the "left team" as $u_{\mathrm{L}}(\boldsymbol{a}) = \sum_{i \in V} u_{i_{\mathrm{L}}}(\boldsymbol{a})$ and the utility of the "right team" as $u_{\mathrm{R}}(a) = \sum_{i \in V} u_{i_{\mathrm{R}}}(a)$. Then $u_{\mathrm{L}}(\boldsymbol{a}) = -u_{\mathrm{R}}(\boldsymbol{a})$.*

*Proof.* The statement follows from straightforward calculations

$$u_{\mathrm{L}}(\boldsymbol{a}) = \sum_{i \in V} u_{i_{\mathrm{L}}}(\boldsymbol{a})$$

$$= \sum_{i \in V} \sum_{j:(i,j) \in E} \tilde{A}^{i_{\mathrm{L}},j_{\mathrm{R}}}(a_{i_{\mathrm{L}}}, a_{j_{\mathrm{R}}})$$

$$= -\sum_{i \in V} \sum_{j:(i,j) \in E} (A^{j,i})^{\top}(a_{i_{\mathrm{L}}}, a_{j_{\mathrm{R}}}) \qquad (\tilde{A}^{i_{\mathrm{L}},j_{\mathrm{R}}} = -(A^{j,i})^{\top})$$

$$= -\sum_{i \in V} \sum_{j:(i,j) \in E} (A^{i,j})^{\top}(a_{j_{\mathrm{L}}}, a_{i_{\mathrm{R}}}) \qquad \text{(By swapping the sum and the identity of } i \text{ and } j)$$

$$= -\sum_{i \in V} \sum_{j:(i,j) \in E} A^{i,j}(a_{i_{\mathrm{R}}}, a_{j_{\mathrm{L}}})$$

$$= -\sum_{i \in V} \sum_{j:(i,j) \in E} \tilde{A}^{i_{\mathrm{R}},j_{\mathrm{L}}}(a_{i_{\mathrm{R}}}, a_{j_{\mathrm{L}}}) \qquad (\tilde{A}^{i_{\mathrm{R}},j_{\mathrm{L}}} = A^{i,j})$$

$$= -\sum_{i \in V} u_{i_{\mathrm{R}}}(\boldsymbol{a})$$

$$= -u_{\mathrm{R}}(\boldsymbol{a}),$$

concluding the proof. $\qquad \square$

**Lemma 3.3.** *Given a $z \in \mathcal{S}^{\nu}$, and $i \in V$, it holds that $\|m_{i_{\mathrm{L}}}(z) - m_{i_{\mathrm{R}}}(z)\|_{\infty} \leq \nu$. Moreover, given a $z \in \Delta(A^{|\mathcal{N}|})$ and an $i \in V$, the set of safe deviations of player $i_{\mathrm{L}}$ is $\Phi_{i_{\mathrm{L}}}^{\mathcal{S}}(z) = \{x_{i_{\mathrm{L}}} : \|x_{i_{\mathrm{L}}} - m_{i_{\mathrm{R}}}(z)\|_{\infty} \leq 0\}$, which guarantees $\Phi_{i_{\mathrm{L}}}^{\mathcal{S}}(z) \neq \emptyset$.*

*Proof.* Consider the case $j \leq k$:

$$C_{i_{\mathrm{L}}}^{j}(z) = \sum_{\boldsymbol{a} \in \mathcal{A}} C_{i_{\mathrm{L}}}^{j}(\boldsymbol{a}) z(\boldsymbol{a})$$

$$= \sum_{\boldsymbol{a} \in A^n : a_{i_{\mathrm{L}}} = j, a_{i_{\mathrm{R}}} \neq j} z(\boldsymbol{a}) - \sum_{\boldsymbol{a} \in A^n : a_{i_{\mathrm{L}}} \neq j, a_{i_{\mathrm{R}}} = j} z(\boldsymbol{a})$$

$$= \sum_{\boldsymbol{a} \in \mathcal{A} : a_{i_{\mathrm{L}}} = j} z(\boldsymbol{a}) - \sum_{\boldsymbol{a} \in \mathcal{A} : a_{i_{\mathrm{R}}} = j} z(\boldsymbol{a})$$

$$= m_{i_{\mathrm{L}}}(z|j) - m_{i_{\mathrm{R}}}(z|j)$$

and thus $C_{i_{\mathrm{L}}}^{j}(z) \leq \nu$, with $j \leq k$ implies that $m_{i_{\mathrm{L}}}(z|j) - m_{i_{\mathrm{R}}}(z|j) \leq \nu$. On the other hand $C_{i_{\mathrm{L}}}^{j}(z) \leq 0$, with $j > k$ implies that $m_{i_{\mathrm{R}}}(z|j) - m_{i_{\mathrm{L}}}(z|j) \leq 0$ which concludes the statement. $\square$

## C   Missing proofs and additional details from Section 4 (membership)

### C.1   Explicit definition of $A(\tilde{z})$

The correspondence $Q : [0,1]^{\ell n} \rightrightarrows [0,1]^{\ell n}$ is given by $Q(\tilde{z}) = \{z : A(\tilde{z})z \leq b(\tilde{z})\}$ where

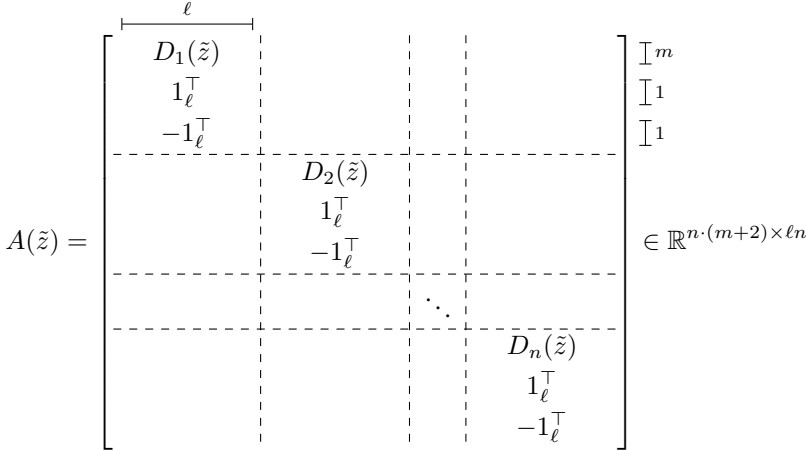

where $D_i(\tilde{z}) \in \mathbb{R}^{m \times \ell}$ is a matrix such that $[0_{m \times \ell}, \cdots, D_i(\tilde{z}), \cdots, 0_{m \times \ell}]z \leq 0_m \iff C_i^{j}(\tilde{p})p_i \leq 0 \, \forall j \in [m]$. In particular, for each $i$, only the components of $z$ corresponding to the strategies of the $i$-th player matter and correspond to the strategy $p_i$. We define $g$ the flattening function which takes a product distribution $\tilde{p} = \bigotimes_{i \in [n]} \tilde{p}_i$ and returns the corresponding "unflatted" vector $\tilde{z}$ while we call $h$ its inverse. Thus, we can write that $D_i(g(\tilde{p}))p_i \leq 0_m$ if and only if $C_i^{j}(p_i \otimes \tilde{p}_{-i}) \leq 0$ for all $j \in [m]$. Notice that $p_i \mapsto C_i^{j}(p_i \otimes \tilde{p}_{-i})$ is linear and can be written as $C_i^{j}(p_i \otimes \tilde{p}_{-i}) = c_i^{j}(\tilde{p})^{\top} p_i$, where $c_i^{j}(\tilde{p}) \in \mathbb{R}^{\ell}$ and each component $c_i^{j}(\tilde{p})_{\bar{a}}$, indexed by $\bar{a}$, is given by $\sum_{\boldsymbol{a} \in \mathcal{A} : a_i = \bar{a}} C_i^{j}(\boldsymbol{a}) \prod_{k \neq i} \tilde{p}_k(a_k)$. Consequently, we can take

$$D_i(\tilde{z}) = \begin{bmatrix} c_i^1(h(\tilde{z})) \\ \vdots \\ c_i^m(h(\tilde{z})) \end{bmatrix} \in \mathbb{R}^{m \times \ell n}$$

### C.2   Proof of Claim 4.3 and Claim 4.4 from Proposition 4.2

**Claim 4.3.** *The functions $\tilde{z} \mapsto A(\tilde{z})z$ and $\tilde{z} \mapsto b(\tilde{z})^{\top} z$ defining the correspondence are L-Lipschitz for every $z \in [0,1]^{\ell n}$ where L has a representation polynomial in the size of the instance.*

*Proof.* First note that $b(\tilde{z})$ does not depend on $\tilde{z}$ and thus is trivially 0-Lipschitz. Thus, we only need to prove the statement about $A(\tilde{z})$. We recall the exact definition of $A(\tilde{z})$ given in Appendix C.1.

We analyze the Jacobian of the $D_i(\tilde{z})$. Any entry of $D_i(\tilde{z})$ would correspond to a cost $j$ of the $i$-th player (rows) and to an action $\hat{a} \in A$ (columns), and any component $\ell$ of $\tilde{z}$ would correspond to a player $i'$ (possibly different from $i$) and an action $\bar{a} \in A$. Thus, by defining $\tilde{p} = h(\tilde{z})$, we can compute the following

$$\frac{\partial c_i^j(h(\tilde{z}))_{\hat{a}}}{\partial \tilde{z}_\ell} = \frac{\partial c_i^j(\tilde{p})_{\hat{a}}}{\partial \tilde{p}_{i'}(\bar{a})} \frac{\partial \tilde{p}_{i'}(\bar{a})}{\partial \tilde{z}_\ell}$$

The second term is clearly 1, as the function $h$ just rearranges the components $\tilde{z}$, while the first term is easily bounded as follows

$$\left| \frac{\partial c_i^j(\tilde{p})_{\hat{a}}}{\partial \tilde{p}_{i'}(\bar{a})} \right| = \left| \frac{\partial \sum_{\boldsymbol{a} \in \mathcal{A}: a_i = \hat{a}} C_i^j(\boldsymbol{a}) \prod_{k \in [n], k \neq i} \tilde{p}_k(a_k)}{\partial \tilde{p}_{i'}(\bar{a})} \right|$$

$$= \left| \sum_{\boldsymbol{a} \in \mathcal{A}: a_{i'} = \bar{a}, a_i = \hat{a}} C_i^j(\boldsymbol{a}) \prod_{k \in [n], k \neq i', k \neq i} \tilde{p}_k(a_k) \right| \leq \ell^n.$$

The following elementary lemma lets us conclude the proof.

**Lemma C.1.** *Let $M : \mathbb{R}^K \to \mathbb{R}^{m \times n}$ be a matrix valued function such that $\left| \frac{\partial M_{i,j}(\tilde{z})}{\partial \tilde{z}_k} \right| \leq C$ for all $i \in [m], j \in [n], k \in [K]$ then*

$$\|(M(\tilde{z}) - M(\tilde{z}'))z\| \leq Cm\sqrt{nK}\|\tilde{z} - \tilde{z}'\|,$$

*for all $z \in [0,1]^K$.*

Indeed,

$$\|(A(\tilde{z}) - A(\tilde{z}'))z\| \leq \ell^n(\ell n)\sqrt{\ell n \cdot n(m + \ell + 2)}\|\tilde{z} - \tilde{z}'\|$$

$$\leq 2\ell^{n+2}n^2\sqrt{m}\|\tilde{z} - \tilde{z}'\|$$

and thus $L = \text{poly}(\ell^n, m, n)$ concluding the proof. $\qquad \square$

**Claim 4.4.** *The operator $F : [0,1]^{\ell n} \to [0,1]^{\ell n}$ is $G$-Lipschitz where $G$ has a representation polynomial in the size of the instance.*

*Proof.* We can get a simple upper bound on the Lipschitz constant of $F$ by bounding its gradient. In particular $F(z) = (-\nabla_{p_1} u_1(p), \ldots, -\nabla_{p_n} u_n(p))$, where as usual $z$ is the unrolling of the product distribution $p = \bigotimes_{i=1}^n p_i$. We can consider any component of $F$, which will correspond to some player $i \in [n]$ and action $\bar{a} \in A$, and consider some component of $z$ which will correspond to some player $j \in [n]$ and some action $\tilde{a} \in A$. The component of $F$ selected corresponds to $-\frac{\partial u_i(p)}{\partial p_i(\bar{a})}$ We can then consider the following:

$$-\frac{\partial^2 u_i(p)}{\partial p_j(\tilde{a}) \partial p_i(\bar{a})} = -\sum_{\boldsymbol{a} \in \mathcal{A}: a_i = \bar{a}, a_j = \tilde{a}} u_i(\boldsymbol{a}) \prod_{k \neq i, j} p_k(a_k)$$

and thus $\left| \frac{\partial^2 u_i(p)}{\partial p_j(\tilde{a}) \partial p_i(\bar{a})} \right| \leq \ell^n$. The mean value theorem trivially concludes the proof:

$$\|F(z) - F(z')\| \leq \|J_F(\xi)\| \cdot \|z - z'\|$$

for some $\xi$ on the segment connecting $z$ and $z'$. Now for any matrix $M \in \mathbb{R}^{m \times n}$ it holds that $\|M\| \leq \sqrt{mn} \cdot \sup_{i,j} |M_{i,j}|$ and thus $\|J_F(\xi)\| \leq n\ell^{n+1} = G$, concluding the proof. $\qquad \square$

**Claim 4.5.** *Let $\nu' = \min(\frac{\epsilon}{2}, \frac{\nu^2}{2n})$ and $\epsilon' = \frac{\epsilon}{2}(1 - n\nu')$, then $p$ is such that*

$$u_i(p) \geq u_i(\tilde{p}_i \otimes p_{-i}) - \epsilon \quad \forall \tilde{p}_i \in \Phi_i^{\mathcal{S}}(p)$$

*and $p \in \mathcal{S}^\nu$.*

*Proof.* First, observe that if $\nu \geq 1$ then the costs bear no effects and PPAD-membership can be established by the PPAD-membership of Nash equilibria. Thus, we can assume w.l.o.g. that $\nu \leq 1$.

**Step 1: Safety** First, we claim that $p$ is $\nu$-safe. Indeed, $\|z^i\|_1 \in [1 - \nu', 1 + \nu']$ and for each $i \in [n], j \in [m]$ and we can directly compute

$$\sum_{\boldsymbol{a} \in A^n} C_i^j(\boldsymbol{a}) \prod_{j \in [n]} z^j(a_j) \leq \nu',$$

then, we can divide the left and right hand side by $\prod_{j \in [n]} \|z^j\|_1 \geq (1 - \nu')^n$ and, obtain that:

$$C_i^j(p) = \sum_{\boldsymbol{a} \in A^n} C_i^j(\boldsymbol{a}) \prod_{j \in [n]} p_j(a_j) \leq \frac{\nu'}{(1 - \nu')^n} \leq \frac{\nu'}{1 - n\nu'} \leq \nu,$$

where in the last inequality we used $\nu' \leq \frac{\nu^2}{2n} \leq \frac{\nu}{1+n\nu}$ for all $\nu \geq 0$ and $n \geq 1$. This shows that indeed $p \in \mathcal{S}^\nu$.

**Step 2: Simulating deviations** Now we prove that we can simulate safe deviations $\tilde{p}_i$ with appropriate choices of $\tilde{z}$.

Take any safe deviation $\tilde{p}_i \in \Delta(A)$ (*i.e.*, it holds that $C_i^j(\tilde{p}_i \otimes p_{-i}) \leq 0$), and consider $\tilde{z}$ defined as $\tilde{z} = [z^1, \ldots, \tilde{z}^i, \ldots, z^n]$, where $\tilde{z}^i = \tilde{p}_i$. By the definition of the correspondence (see Appendix C.1 for notation and details on the explicit construction of the correspondence $Q$) we have that $\tilde{z} \in Q_{\nu'}(z)$. Indeed, from the definition of $D_i$ in Appendix C.1, we get that

$$
\begin{aligned}
[D_i(z)\tilde{z}^i]_j &= c_i^j(h(z))^\top \tilde{z}^i && \text{(by construction)} \\
&= \sum_{\bar{a} \in A} \sum_{\boldsymbol{a} \in A^n, a_i = \bar{a}} C_i^j(\boldsymbol{a}) \prod_{k \neq i} z^k(a_k) \tilde{z}^i(\bar{a}) && \text{(def. of } c_i^j(h(z))) \\
&= \sum_{\boldsymbol{a} \in A^n, a_i} C_i^j(\boldsymbol{a}) \prod_{k \neq i} z^k(a_k) \tilde{z}^i(a_i) && \\
&\leq 0 && (C_i^j(\tilde{p}_i \otimes p_{-i}) \leq 0)
\end{aligned}
$$

Moreover, it is easy to verify that $1^\top \tilde{z}^i \in [1 - \nu', 1 + \nu']$ for all $i \in [n]$. This shows that $\tilde{z} \in Q_{\nu'}(z)$.

Formally, what we proved is that: for any player $i \in [n]$ and for any safe deviation $\tilde{p}_i$ such that $C_i^j(\tilde{p}_i \otimes p_{-i}) \leq 0$, there exists $\tilde{z} \in Q_{\nu'}(z)$ which "simulates" $\tilde{p}_i$. Moreover, $\tilde{z}^j = z^j$ for all $j \neq i$ and $\tilde{z}^i = \tilde{p}_i$.

**Step 3: Equilibrium conditions** Now, we also claim that $p = \bigotimes_{i \in [n]} p_i$ satisfies the equilibrium constraints. Since $z$ is a solution to QUASIVI we have that for all $\tilde{z} \in Q_{\nu'}(z)$ the following holds:

$$F(z)^\top (\tilde{z} - z) \geq -\epsilon' \implies -\sum_{i \in [n]} \sum_{\bar{a} \in A} \sum_{\boldsymbol{a} \in \mathcal{A}: a_i = \bar{a}} u_i(\boldsymbol{a}) \prod_{j \neq i} z^j(a_j)(\tilde{z}^i(\bar{a}) - z^i(\bar{a})) \geq -\epsilon',$$

which implies that:

$$\sum_{\boldsymbol{a} \in \mathcal{A}} u_i(\boldsymbol{a}) \prod_{j \neq i} z^j(a_j)(z^i(a_i) - \tilde{z}^i(a_i)) \geq -\epsilon',$$

once we specialize to $\tilde{z}$ to the one built in step 2, since $\tilde{z}^j = z^j$ for all $j \neq i$.

Now we can use the exact definition of $\tilde{z}^i$ constructed in step 2, in the above equation, which leads to the following:

$$\sum_{\boldsymbol{a} \in \mathcal{A}} u_i(\boldsymbol{a}) \prod_{j \neq i} z^j(a_j)(z^i(a_i) - \tilde{z}^i(a_i)) = \sum_{\boldsymbol{a} \in \mathcal{A}} u_i(\boldsymbol{a}) \prod_{j \in [n]} z^j(a_j) - \sum_{\boldsymbol{a} \in \mathcal{A}} u_i(\boldsymbol{a}) \prod_{j \neq i} z^j(a_j) \tilde{p}_i(a_i) \geq -\epsilon'.$$

Rearranging it, we obtain

$$\sum_{\boldsymbol{a} \in \mathcal{A}} u_i(\boldsymbol{a}) \prod_{j \in [n]} z^j(a_j) \geq \sum_{\boldsymbol{a} \in \mathcal{A}} u_i(\boldsymbol{a}) \prod_{j \neq i} z^j(a_j) \tilde{p}_i(a_i) - \epsilon'.$$

We can now divide each term by $\gamma = \prod_{j \in [n]} \|z^j\|_1$, which finally leads to

$$u_i(p) \geq \frac{u_i(\tilde{p}_i \otimes p_{-i})}{\|z^i\|_1} - \frac{\epsilon'}{\gamma} \geq u_i(\tilde{p}_i \otimes p_{-i}) - \left(\frac{\epsilon'}{\gamma} + \nu'\right)$$

and since $\|z^j\|_1 \in [1 - \nu', 1 + \nu']$ for all $j \in [n]$ this implies that $\gamma \geq (1 - \nu')^n \geq 1 - n\nu'$ and thus $u_i(p) \geq u_i(\tilde{p}_i \otimes p_{-i}) - \left(\frac{\epsilon'}{1-n\nu'} + \nu'\right)$.

Consider the first term $\frac{\epsilon'}{1-n\nu'}$. The following inequalities hold:

$$\frac{\epsilon'}{1 - n\nu'} \leq \frac{\epsilon}{2}$$

since $\epsilon' \leq \frac{\epsilon}{2}(1 - n\nu')$.

Moreover, $\nu' \leq \frac{\epsilon}{2}$ thus proving that $u_i(p) \geq u_i(\tilde{p}_i \otimes p_{-i}) - \epsilon$ as desired. $\qquad\square$

## C.3 Additional technical lemmas

**Lemma C.1.** *Let $M : \mathbb{R}^K \to \mathbb{R}^{m \times n}$ be a matrix valued function such that $\left|\frac{\partial M_{i,j}(\tilde{z})}{\partial \tilde{z}_k}\right| \leq C$ for all $i \in [m], j \in [n], k \in [K]$ then*

$$\|(M(\tilde{z}) - M(\tilde{z}'))z\| \leq Cm\sqrt{nK}\|\tilde{z} - \tilde{z}'\|,$$

*for all $z \in [0,1]^K$.*

*Proof.* Let $\{m_i : [0,1]^K \to \mathbb{R}^n\}_{i=1}^m$ be the functions defining the rows of $M$ and $h_i(\tilde{z}|z) = m_i(\tilde{z})^\top z$. With this notation it is easy to check that $\nabla_{\tilde{z}} h_i(\tilde{z}|z) = J_{m_i}(\tilde{z})^\top z$ and thus $\|\nabla_{\tilde{z}} h_i(\tilde{z}|z)\| \leq \|J_{m_i}(\tilde{z})\|\|z\| \leq C\sqrt{mnK}$.

By the mean value theorem, we have that for some $\xi$ in the segment connecting $\tilde{z}$ and $\tilde{z}'$, we have

$$|(m_i(\tilde{z}) - m_i(\tilde{z}'))^\top z| \leq \|\nabla_{\tilde{z}} h(\xi|z)\| \cdot \|\tilde{z} - \tilde{z}'\|$$
$$\leq C\sqrt{mnK}\|\tilde{z} - \tilde{z}'\|$$

Finally,

$$\|(M(\tilde{z}) - M(\tilde{z}'))z\|^2 = \sum_{i=1}^m ((m_i(\tilde{z}) - m_i(\tilde{z}'))^\top z)^2$$
$$\leq C^2 m^2 nK\|\tilde{z} - \tilde{z}'\|^2$$

concluding the proof. $\qquad\square$

