# OpenReview forum: "The Complexity of Correlated Equilibria in Generalized Games"
_NeurIPS.cc/2025/Conference — NeurIPS 2025 poster_

### Official Review · Reviewer_BkuP · 2025-06-30

**Clarity:** 2
**Significance:** 3
**Originality:** 3
**Rating:** 4
**Confidence:** 3

**Summary:**

The paper studies the computational complexity of finding certain constrained approximate $\Phi$-equilibria in generalized games. It is shown that this search problem is PPAD-complete and thus presumably not solvable in polynomial time. This answers a recent open question from the literature. PPAD-hardness is based on a reduction from the PolyMatrix problem (finding approximate equilibria in polymatrix games) and membership in PPAD is shown via reduction to some problem related to quasi-variational inequalities.

**Questions:**

1. Since your problem is formulated as a promise problem, do you not need to show for your construction that it satisfies the promise?

2. For the correctness, you only show one direction. Is this enough because we consider total search problems (where existence of a solution is guaranteed)?

3. Are there any interesting special cases which are efficiently solvable?

**Ethical Concerns:**

["NO or VERY MINOR ethics concerns only"]

**Final Justification:**

The paper resolves a relevant theoretical open question. Given that the authors discuss the practical relevance and implications of their result, I find it acceptable.

**Limitations:**

No limitations are explicitly mentioned.

**Paper Formatting Concerns:**

None.

**Quality:**

3

**Strengths And Weaknesses:**

Strengths: The paper settles the complexity status of a seemingly important search problem in (generalized) game theory.

Weakness:
- The result is purely theoretic and I do not see a direct connection to the NeurIPS community. There is no discussion on possible practical implications of the result.
- The presentation is not understandable for non-experts in generalized games. Many important notions are not properly explained and defined (e.g. What is $\Delta(\mathcal A)$ and $\Delta(A)$? What is the idea behind the mediator recommendation and deviation? What is correlated distribution?).
- From a theoretical perspective, since I am not an expert in the field, I am not sure how relevant or surprising the result is.
- The paper misses a concluding discussion (other directions or open cases).

---

> ### Author Rebuttal · Authors · 2025-07-31
>
> We thank the reviewer for their comments and suggestions. The response is divided into two parts: one common section addressing general concerns, followed by answers to specific questions.
>
> ***
>
> We identified the following two points as the main concerns among the reviewers: 1) relevance to the NeurIPS community and 2) the lack of practical consequences of an hardness result. In the following, we address these two points.
>
> ## Relevance
>
> First, we emphasize that convergence‐to‐equilibrium results in multi‐agent learning occupy a central place in the NeurIPS community and in the broader ML community. Our work delivers a fundamentally negative insight: it rules out the existence of efficient algorithms for computing constrained correlated equilibria, in contrast to wide array of algorithms available in the unconstrained case. In particular, there are many recent works on no-regret algorithms for correlated equilibria, and one could expect the same kind of tools to be applicable in the constrained setting. In contrast, we show that efforts to establish positive convergence guarantees of the form “no-regret algorithms converge to constrained correlated equilibria” are going to fail. We believe it is important to broadcast this impossibility result: by doing so, we can steer the community away from unfruitful directions and toward questions where real progress can be made. The NeurIPS community is precisely where researchers most invested in proving positive convergence results gather, making our impossibility result both timely and highly relevant to their efforts.
>
> We also highlight that our results answer a known open problem within the ML community (see [1,2]). Prior work explicitly left open the complexity of finding these equilibria. We settle this question definitively in the negative. This hardness result fills a clear gap in the theory: while numerous works have established positive convergence rates for regret minimization in unconstrained settings, the constrained case remained largely open until now.
>
> Finally, it is not new for leading ML venues to publish papers whose main contribution is a hardness result, rather than a positive algorithmic guarantee. This is particularly the case in areas close to game theory. As an example, we mention the recent paper “The Complexity of Two-Team Polymatrix Games with Independent Adversaries” by Hollender et al. [3] that was very well received at ICLR 25.
>
> ## Practical relevance
>
> While focusing on a general and abstract framework, our model captures important applications in the Internet economy, such as automated bidding systems that power modern digital advertising. Today’s platforms typically deploy autobidders that act on behalf of advertisers during the real-time bidding process. Each autobidder dynamically adjusts bids and usually has to satisfy constraints specified by the advertiser, such as return‐on‐investment, budget, and cost‑per‑acquisition constraints. These autobidding systems have attracted extensive attention in machine learning, originating a rich body of both theoretical and empirical works (see, e.g., [4,5,6,7]).
>
> One key aspect of these systems is that all autobidders are deployed and run directly by the advertising platform. Therefore, platforms are usually interested in designing autobidding algorithms that converge to equilibrium over time, as this makes the system more stable, predictable, and compatible with the individual incentives of each advertiser. Our impossibility result demonstrates that, in general, convergence cannot be guaranteed. Consequently, to design autobidding platforms with provable convergence guarantees, advertising systems must employ new techniques that exploit the specific structure of their problem, since no general-purpose algorithm will suffice.
>
> Due to space constraints, this point was only briefly mentioned in the current version. We will elaborate on it in the final version of the paper.
>
>
> ## References
>
> [1] Zhang, B. H., Anagnostides, I., Tewolde, E., Berker, R. E., Farina, G., Conitzer, V., & Sandholm, T. (2025). Expected variational inequalities. ICML. 2025.
>
> [2] Bernasconi, M., Castiglioni, M., Marchesi, A., Trovo, F., & Gatti, N. (2023, July). Constrained phi-equilibria. In International Conference on Machine Learning (pp. 2184-2205). PMLR.
>
> [3] Hollender, Alexandros, Gilbert Maystre, and Sai Ganesh Nagarajan. "The Complexity of Two-Team Polymatrix Games with Independent Adversaries." ICLR. 2025.
>
> [4] Balseiro, Santiago R., and Yonatan Gur. "Learning in Repeated Auctions with Budgets: Regret Minimization and Equilibrium." Proceedings of the 2017 ACM Conference on Economics and Computation. 2017.
>
> [5] Paes Leme, R., Piliouras, G., Schneider, J., Spendlove, K., & Zuo, S. (2024, July). Complex dynamics in autobidding systems. In Proceedings of the 25th ACM Conference on Economics and Computation (pp. 75-100).
>
> [6] Balseiro, S., Deng, Y., Mao, J., Mirrokni, V., & Zuo, S. (2021). Robust auction design in the auto-bidding world. Advances in Neural Information Processing Systems, 34, 17777-17788.
>
> [7] Agarwal, D., Ghosh, S., Wei, K., & You, S. (2014, August). Budget pacing for targeted online advertisements at linkedin. In Proceedings of the 20th ACM SIGKDD international conference on Knowledge discovery and data mining (pp. 1613-1619).
>
> ***
>
> ## Specific answers
>
> We are sorry if the paper was outside of the area of expertise of the reviewer. Nonetheless, we will address the reviewer’s questions thoroughly in the hope that this can help to better reevaluate the significance of our work.
>
> > The result is purely theoretic..
>
> We refer to the common answer for the practical relevance of our results and their interest in the NeurIPS community.
>
> > The presentation is not understandable for non-experts...
>
> Due to space constraints, we had to cut the explanation of the most standard game-theory concepts and results. For the reviewer's convenience, we will summarize it here. In particular, $\Delta(A)$, for a finite set $A$ is the set of all categorical distributions on the elements of $A$. A correlated distribution is a distribution over actions that is correlated between the actions of different players, ie, cannot be written as the product of distributions over the actions of single players. In a correlated equilibrium, the object of study is exactly a correlated distribution with some additional properties. Meaning that, when the actions are drawn from the correlated distribution and communicated privately to each player, they (employing Bayesian reasoning on the distribution of play of the other player, after having seen their recommendation), are not incentivised to deviate from the recommendation.
>
> ## Answers to the questions
>
> > 1. Since your problem is formulated as a promise problem...
>
> Yes, indeed. The only promise that needs to be proved is that the set of deviations is non-empty. This, although not explicitly highlighted, is proven in Lemma 3.3.
>
> > 2. For the correctness, you only show one direction...
>
> You are right; reductions for TFNP problems behave differently than reductions between NP problems. This is because we only have Yes instances by definition of totality.
>
> > 3. Are there any interesting special cases which are efficiently solvable?
>
> The cases in which efficient algorithms can be found are discussed in prior works (which left open the question on the computational complexity of finding any equilibria, which we address here). We refer to the answer to review dM7m (answer c) for more details on this.

---

> > ### Comment · Reviewer_BkuP · 2025-08-04
> >
> > Thank you for your reponse. I will raise my score to 4.

---

> > > ### Author Response · Authors · 2025-08-04
> > >
> > > We appreciate the reviewer’s time and effort in reevaluating our work

---

### Official Review · Reviewer_KuZX · 2025-06-30

**Clarity:** 3
**Significance:** 2
**Originality:** 2
**Rating:** 3
**Confidence:** 3

**Summary:**

The paper studies the notion of \Phi-correlated equilibrium in n player games under cost constraints for agents which are emposed on the their joint actions.
More precisely, the set \Phi describes the set of (possibly randomized) deviations each player can make from action a for each action a. The equilibrium notion then restricts the players to use these deviations to deviate from their recommended strategy which is sampled by a mediator who correlates the agents joint actions.

Agents should not want to use any of the allowed deviations.
Moreover, each agent has an m dimensional cost vector associated with each joint action profile. For each cost dimension each agent has a threshold cost which she does not want to cross in expectation.
If this holds for all cost dimensions the correlated action profile is considered safe and one can define the induced set of safe deviations naturally.

Because safe deviations may not always exist, the authors constrain to a setting where each agent has at least one safe deviation - that is we are promised that an equilibrium exists.

The paper then shows that finding a constrained \Phi-correlated equilibrium is PPAD-complete.

**Questions:**

You mention some applications in the introduction such as repeated auctions/ online advertising. Can you explain how your specific hardness result is relevant for the specific applications you mention?

Which sets of functions \Phi are relevant other than CCE and CE?
Can you name some applications in which other restricted type of deviations would only be feasible?

**Ethical Concerns:**

["NO or VERY MINOR ethics concerns only"]

**Final Justification:**

The paper is borderline for NeurIPS, the result is good to have, but not surprising and the result does not answer the specific practical motivation mentioned in the introduction.

**Limitations:**

Lack of directly relevant applications

**Quality:**

3

**Strengths And Weaknesses:**

The paper is well written, although at times a bit verbose (e.g. 67-69).
Overall, I think the paper is fine, but do not see the usefulness of the result and further it doesn't seem technically
challenging enough to accept it despite this.

Although not my main concern, it is not clear why \Phi equilibrium in its generality is useful, especially since the deviation agents are only not incentivized to deviate in an average sense.

__

Minor errors:

In Line 201, in the second line and in Line 212 i and j seem to be flipped

---

> ### Author Rebuttal · Authors · 2025-07-31
>
> We thank the reviewer for their comments and suggestions. The response is divided into two parts: one common section addressing general concerns, followed by answers to specific questions.
>
> ***
>
> We identified the following two points as the main concerns among the reviewers: 1) relevance to the NeurIPS community and 2) the lack of practical consequences of an hardness result. In the following, we address these two points.
>
> ## Relevance
>
> First, we emphasize that convergence‐to‐equilibrium results in multi‐agent learning occupy a central place in the NeurIPS community and in the broader ML community. Our work delivers a fundamentally negative insight: it rules out the existence of efficient algorithms for computing constrained correlated equilibria, in contrast to wide array of algorithms available in the unconstrained case. In particular, there are many recent works on no-regret algorithms for correlated equilibria, and one could expect the same kind of tools to be applicable in the constrained setting. In contrast, we show that efforts to establish positive convergence guarantees of the form “no-regret algorithms converge to constrained correlated equilibria” are going to fail. We believe it is important to broadcast this impossibility result: by doing so, we can steer the community away from unfruitful directions and toward questions where real progress can be made. The NeurIPS community is precisely where researchers most invested in proving positive convergence results gather, making our impossibility result both timely and highly relevant to their efforts.
>
> We also highlight that our results answer a known open problem within the ML community (see [1,2]). Prior work explicitly left open the complexity of finding these equilibria. We settle this question definitively in the negative. This hardness result fills a clear gap in the theory: while numerous works have established positive convergence rates for regret minimization in unconstrained settings, the constrained case remained largely open until now.
>
> Finally, it is not new for leading ML venues to publish papers whose main contribution is a hardness result, rather than a positive algorithmic guarantee. This is particularly the case in areas close to game theory. As an example, we mention the recent paper “The Complexity of Two-Team Polymatrix Games with Independent Adversaries” by Hollender et al. [3] that was very well received at ICLR 25.
>
> ## Practical relevance
>
> While focusing on a general and abstract framework, our model captures important applications in the Internet economy, such as automated bidding systems that power modern digital advertising. Today’s platforms typically deploy autobidders that act on behalf of advertisers during the real-time bidding process. Each autobidder dynamically adjusts bids and usually has to satisfy constraints specified by the advertiser, such as return‐on‐investment, budget, and cost‑per‑acquisition constraints. These autobidding systems have attracted extensive attention in machine learning, originating a rich body of both theoretical and empirical works (see, e.g., [4,5,6,7]).
>
> One key aspect of these systems is that all autobidders are deployed and run directly by the advertising platform. Therefore, platforms are usually interested in designing autobidding algorithms that converge to equilibrium over time, as this makes the system more stable, predictable, and compatible with the individual incentives of each advertiser. Our impossibility result demonstrates that, in general, convergence cannot be guaranteed. Consequently, to design autobidding platforms with provable convergence guarantees, advertising systems must employ new techniques that exploit the specific structure of their problem, since no general-purpose algorithm will suffice.
>
> Due to space constraints, this point was only briefly mentioned in the current version. We will elaborate on it in the final version of the paper.
>
>
> ## References
>
> [1] Zhang, B. H., Anagnostides, I., Tewolde, E., Berker, R. E., Farina, G., Conitzer, V., & Sandholm, T. (2025). Expected variational inequalities. ICML. 2025.
>
> [2] Bernasconi, M., Castiglioni, M., Marchesi, A., Trovo, F., & Gatti, N. (2023, July). Constrained phi-equilibria. In International Conference on Machine Learning (pp. 2184-2205). PMLR.
>
> [3] Hollender, Alexandros, Gilbert Maystre, and Sai Ganesh Nagarajan. "The Complexity of Two-Team Polymatrix Games with Independent Adversaries." ICLR. 2025.
>
> [4] Balseiro, Santiago R., and Yonatan Gur. "Learning in Repeated Auctions with Budgets: Regret Minimization and Equilibrium." Proceedings of the 2017 ACM Conference on Economics and Computation. 2017.
>
> [5] Paes Leme, R., Piliouras, G., Schneider, J., Spendlove, K., & Zuo, S. (2024, July). Complex dynamics in autobidding systems. In Proceedings of the 25th ACM Conference on Economics and Computation (pp. 75-100).
>
> [6] Balseiro, S., Deng, Y., Mao, J., Mirrokni, V., & Zuo, S. (2021). Robust auction design in the auto-bidding world. Advances in Neural Information Processing Systems, 34, 17777-17788.
>
> [7] Agarwal, D., Ghosh, S., Wei, K., & You, S. (2014, August). Budget pacing for targeted online advertisements at linkedin. In Proceedings of the 20th ACM SIGKDD international conference on Knowledge discovery and data mining (pp. 1613-1619).
>
> ***
>
> ## Specific answers
>
> > You mention some applications in the introduction...
>
> For the relevance of our result, please refer to the common answer.
>
> > Which sets of functions $\Phi$ are relevant...
>
> Phi equilibria are an important generalization of equilibria in games, thanks to their deep connection to the level of rationality of the players, and they were pivotal in addressing important results in game theory (for instance [8]). Indeed, Phi deviations not only generalize both CEs and CCEs but also allow for a much more general type of rationality. For example, we could easily envision a setting in which deviation functions are all the possible ones for most of the actions, but when the player is recommended some critical actions, they are not allowed to deviate due, for instance, to safety constraints. Another motivation for considering general sets of deviations is that different sets may result in different properties in terms of manipulability (see, for instance, [9] and references therein).
>
> [8] Celli, Andrea, et al. "No-regret learning dynamics for extensive-form correlated equilibrium." Advances in Neural Information Processing Systems 33 (2020): 7722-7732.
>
> [9] Arunachaleswaran, E. R., Collina, N., Mansour, Y., Mohri, M., Schneider, J., & Sivan, B. (2025, July). Swap Regret and Correlated Equilibria Beyond Normal-Form Games. In Proceedings of the 26th ACM Conference on Economics and Computation (pp. 130-157).

---

> ### Author Response · Authors · 2025-08-06
>
> Thank you again for your helpful feedback. As the discussion period ends in less than two days, we’d appreciate it if you could let us know whether our responses address your concerns regarding the usefulness of the result and the lack of applications. We’re happy to clarify anything further if needed.

---

> > ### Comment · Reviewer_KuZX · 2025-08-06
> >
> > Thank you for your response!
> > Do you have more examples pure PPAD hardness papers at ML conferences than the one you mentioned? I am aware of it, but think of it as an exception rather than the rule.
> >
> > Regarding the open problem you solve, I would change the reference in Line 44 in your paper to be [2] not [1], as this is the paper in which the problem is actually listed in the final section.
> >
> > You mention autobidding as a motivation and explain why constrains can make sense in some settings, which is reasonable. Then you show that A) in general, if you have constraints, you cannot compute CEs efficiently and you don't show B) in the autobidding problem you cannot compute CEs efficiently. I believe showing B would be much more interesting for the ML community, namely showing it is hard the specific application instead of showing that a very general version of the problem is hard.
> >
> > [1] Zhang, B. H., Anagnostides, I., Tewolde, E., Berker, R. E., Farina, G., Conitzer, V., & Sandholm, T. (2025). Expected variational inequalities. ICML. 2025.
> >
> > [2] Bernasconi, M., Castiglioni, M., Marchesi, A., Trovo, F., & Gatti, N. (2023, July). Constrained phi-equilibria. In International Conference on Machine Learning (pp. 2184-2205). PMLR.

---

> ### Author Response · Authors · 2025-08-07
>
> We thank the reviewer for engaging in the discussion.
>
> We mentioned [1] as an example because it was very well received, recent, and related to our own work, but there are further examples of such hardness results at ML conferences, and we should have mentioned even these in the rebuttal. For example, [2], which studies a topic very related to ours (i.e., the (non) existence of regret minimization approaches to compute correlated equilibria in Markov games), [3] which also studies the complexity of equilibrium computation problems, and finally [4] in which the question is the hardness of mean field games.
>
> On your second point, we agree with the reviewer that our result shows hardness in “general” and not on instances specifically arising from the example. However, we can raise two different points in this regard. First, the properties of the instances arising from our reduction already exclude many possible algorithmic approaches to computing equilibria in bidding scenarios. For example, one possible approach to the autobidding problem could be considering settings with only a small number of bids available in order to simplify the bidders’ action space. However, our results already rule out this kind of approach, since in our instances each agent has 2 actions, which could be interpreted as “bid high”/ “bid low”. This suggests that restricting the action space of the learners is not the “right” additional structure that should be exploited to obtain positive algorithmic results for the autobidding problem.
> Second, it is indeed not trivial to study specific cases of reductions; clearly, the more specific you want your instance, the harder the reduction is. It is common to first have reductions that are quite general and later specify them for a specific problem of interest. For example, [2] motivates the setting by presenting important applications of Markov games, but the instances produced by their reduction do not fit directly any known application.
>
> However, we agree with the reviewer that this is an extremely interesting future line of research, which should be mentioned explicitly in the final version.
>
>
> [1] Hollender, Alexandros, Gilbert Maystre, and Sai Ganesh Nagarajan. "The Complexity of Two-Team Polymatrix Games with Independent Adversaries." ICLR 2025.
>
> [2] Foster, Dylan J., Noah Golowich, and Sham M. Kakade. "Hardness of independent learning and sparse equilibrium computation in markov games." ICML 2023.
>
> [3] Daskalakis, Constantinos, Noah Golowich, and Kaiqing Zhang. "The complexity of markov equilibrium in stochastic games." COLT 2023.
>
> [4] Yardim, Batuhan, Artur Goldman, and Niao He. "When is Mean-Field Reinforcement Learning Tractable and Relevant?." AAMAS 2024.

---

### Official Review · Reviewer_dM7m · 2025-07-02

**Clarity:** 3
**Significance:** 3
**Originality:** 3
**Rating:** 4
**Confidence:** 3

**Summary:**

This paper studies the complexity of computing correlated equilibria (CE) in generalized games, where players’ strategies are interdependent. While CE is typically easier to compute than Nash equilibria, the authors show that in these settings with coupled constraints, even to approximate CE computation is PPAD-complete. The authors establish it via a reduction from polymatrix games using two-team zero-sum game constructions. The results hold even with small approximation or constraint violations. Under standard assumptions, they further show that the problem likely requires quasi-polynomial time, resolving a key open question. This is a great paper to read overall.

**Questions:**

Please answer the following questions:

(a) This paper presents deep theoretical insights but lacks experiments or simulations to illustrate practical implications... such as how these hardness results manifest in real-world economic or multi-agent systems? Can you highlight one or two real-life usecases wherein these results are of significant use?

(b) The hardness proofs rely on structured reductions (eg: from polymatrix games) and certain types of deviation sets.. thus, I believe that these results may not generalize easily to broader classes of games as well as different equilibrium refinements... if not, can you please elaborate more on this?

(c) The results in this paper only highlight worst-case intractability.. they do not explore whether practical heuristics or approximation schemes may still be effective in realistic settings (and thereby offering limited guidance for algorithmic design in applied domains)?

**Ethical Concerns:**

["NO or VERY MINOR ethics concerns only"]

**Limitations:**

(a) This paper presents deep theoretical insights but lacks experiments or simulations to illustrate practical implications... such as how these hardness results manifest in real-world economic or multi-agent systems? Can you highlight one or two real-life usecases wherein these results are of significant use?

(b) The hardness proofs rely on structured reductions (eg: from polymatrix games) and certain types of deviation sets.. thus, I believe that these results may not generalize easily to broader classes of games as well as different equilibrium refinements... if not, can you please elaborate more on this?

(c) The results in this paper only highlight worst-case intractability.. they do not explore whether practical heuristics or approximation schemes may still be effective in realistic settings (and thereby offering limited guidance for algorithmic design in applied domains)?

**Quality:**

3

**Strengths And Weaknesses:**

Below are my comments on this paper:

(a) This paper presents deep theoretical insights but lacks experiments or simulations to illustrate practical implications... such as how these hardness results manifest in real-world economic or multi-agent systems? Can you highlight one or two real-life usecases wherein these results are of significant use?

(b) The hardness proofs rely on structured reductions (eg: from polymatrix games) and certain types of deviation sets.. thus, I believe that these results may not generalize easily to broader classes of games as well as different equilibrium refinements... if not, can you please elaborate more on this?

(c) The results in this paper only highlight worst-case intractability.. they do not explore whether practical heuristics or approximation schemes may still be effective in realistic settings (and thereby offering limited guidance for algorithmic design in applied domains)?

---

> ### Author Rebuttal · Authors · 2025-07-31
>
> We thank the reviewer for their comments and suggestions. The response is divided into two parts: one common section addressing general concerns, followed by answers to specific questions.
>
> ***
>
> We identified the following two points as the main concerns among the reviewers: 1) relevance to the NeurIPS community and 2) the lack of practical consequences of an hardness result. In the following, we address these two points.
>
> ## Relevance
>
> First, we emphasize that convergence‐to‐equilibrium results in multi‐agent learning occupy a central place in the NeurIPS community and in the broader ML community. Our work delivers a fundamentally negative insight: it rules out the existence of efficient algorithms for computing constrained correlated equilibria, in contrast to wide array of algorithms available in the unconstrained case. In particular, there are many recent works on no-regret algorithms for correlated equilibria, and one could expect the same kind of tools to be applicable in the constrained setting. In contrast, we show that efforts to establish positive convergence guarantees of the form “no-regret algorithms converge to constrained correlated equilibria” are going to fail. We believe it is important to broadcast this impossibility result: by doing so, we can steer the community away from unfruitful directions and toward questions where real progress can be made. The NeurIPS community is precisely where researchers most invested in proving positive convergence results gather, making our impossibility result both timely and highly relevant to their efforts.
>
> We also highlight that our results answer a known open problem within the ML community (see [1,2]). Prior work explicitly left open the complexity of finding these equilibria. We settle this question definitively in the negative. This hardness result fills a clear gap in the theory: while numerous works have established positive convergence rates for regret minimization in unconstrained settings, the constrained case remained largely open until now.
>
> Finally, it is not new for leading ML venues to publish papers whose main contribution is a hardness result, rather than a positive algorithmic guarantee. This is particularly the case in areas close to game theory. As an example, we mention the recent paper “The Complexity of Two-Team Polymatrix Games with Independent Adversaries” by Hollender et al. [3] that was very well received at ICLR 25.
>
> ## Practical relevance
>
> While focusing on a general and abstract framework, our model captures important applications in the Internet economy, such as automated bidding systems that power modern digital advertising. Today’s platforms typically deploy autobidders that act on behalf of advertisers during the real-time bidding process. Each autobidder dynamically adjusts bids and usually has to satisfy constraints specified by the advertiser, such as return‐on‐investment, budget, and cost‑per‑acquisition constraints. These autobidding systems have attracted extensive attention in machine learning, originating a rich body of both theoretical and empirical works (see, e.g., [4,5,6,7]).
>
> One key aspect of these systems is that all autobidders are deployed and run directly by the advertising platform. Therefore, platforms are usually interested in designing autobidding algorithms that converge to equilibrium over time, as this makes the system more stable, predictable, and compatible with the individual incentives of each advertiser. Our impossibility result demonstrates that, in general, convergence cannot be guaranteed. Consequently, to design autobidding platforms with provable convergence guarantees, advertising systems must employ new techniques that exploit the specific structure of their problem, since no general-purpose algorithm will suffice.
>
> Due to space constraints, this point was only briefly mentioned in the current version. We will elaborate on it in the final version of the paper.
>
>
> ## References
>
> [1] Zhang, B. H., Anagnostides, I., Tewolde, E., Berker, R. E., Farina, G., Conitzer, V., & Sandholm, T. (2025). Expected variational inequalities. ICML. 2025.
>
> [2] Bernasconi, M., Castiglioni, M., Marchesi, A., Trovo, F., & Gatti, N. (2023, July). Constrained phi-equilibria. In International Conference on Machine Learning (pp. 2184-2205). PMLR.
>
> [3] Hollender, Alexandros, Gilbert Maystre, and Sai Ganesh Nagarajan. "The Complexity of Two-Team Polymatrix Games with Independent Adversaries." ICLR. 2025.
>
> [4] Balseiro, Santiago R., and Yonatan Gur. "Learning in Repeated Auctions with Budgets: Regret Minimization and Equilibrium." Proceedings of the 2017 ACM Conference on Economics and Computation. 2017.
>
> [5] Paes Leme, R., Piliouras, G., Schneider, J., Spendlove, K., & Zuo, S. (2024, July). Complex dynamics in autobidding systems. In Proceedings of the 25th ACM Conference on Economics and Computation (pp. 75-100).
>
> [6] Balseiro, S., Deng, Y., Mao, J., Mirrokni, V., & Zuo, S. (2021). Robust auction design in the auto-bidding world. Advances in Neural Information Processing Systems, 34, 17777-17788.
>
> [7] Agarwal, D., Ghosh, S., Wei, K., & You, S. (2014, August). Budget pacing for targeted online advertisements at linkedin. In Proceedings of the 20th ACM SIGKDD international conference on Knowledge discovery and data mining (pp. 1613-1619).
>
> ***
>
> ## Specific answers
>
> > (a) This paper presents deep theoretical insights...
>
> For the practical consequences of our result, please refer to the common answer.
>
> > (b) The hardness proofs rely on structured reductions
>
> We point out that our reduction reduces both from polymatrix games but also works when applied to bimatrix games, which are very natural. Also, the set of deviations is the set of CCE deviations. Thus, on both these aspects, we view reduction to produce very natural instances.
> Nonetheless, reductions, by their nature, rely on carefully built instances. They are intended to serve as a guide to what results are possible to achieve. We prove that, in general, one cannot find polynomial-time algorithms, so if one is interested in giving efficient algorithms for some more specialized problem, they **must** exploit some fundamental characteristic of their problem.
>
> > (c) The results in this paper only highlight worst-case intractability...
>
> We believe that prior work has already satisfactorily addressed the positive result in this setting. In particular, Bernasconi, et al., [2023], showed that there exists a quasi-polynomial time algorithm in general (running in $O(n^{\log(n)})$ time for constant approximations which we prove to be tight) and polynomial time algorithms in the case in which the number of actions is small or in the case in which the number of constraints is small.

---

### Official Review · Reviewer_QrtF · 2025-07-06

**Clarity:** 3
**Significance:** 3
**Originality:** 3
**Rating:** 5
**Confidence:** 2

**Summary:**

This paper studies the computational complexity of computing Correlated constrained equilibria in generalized games. Previously, it was shown that constrained-phi-equilibria are PPAD-complete to compute in generalized games, but this kept open the question of whether the introduction of correlation could be a sufficient relaxation to make the problem poly-time computable. This paper resolves this question, showing that correlation is insufficient. The proof proceeds by a reduction to Nash Equilibrium computation in polymatrix games. The result also implies that the quasi-polynomial time algorithm for computing approximate phi-equilibria of Bernasconi et al. [2023] is tight.

**Questions:**

- What are the practical consequences of this work?
- Are there other relaxations of constrained equilibria that may be easy to compute?

**Ethical Concerns:**

["NO or VERY MINOR ethics concerns only"]

**Limitations:**

Yes

**Quality:**

3

**Strengths And Weaknesses:**

Strengths:
- Resolves a fundamental problem in the complexity of computing equilibria for game theoretic problems, an active and important area of research
- Proof is technically involved
- Has broad consequences

Weaknesses:
- Unclear what the practical consequences are -- are there practical settings where the fact that these equilibria are hard to compute may cause issues? Since this paper has been submitted to NeurIPS, it would be nice to talk about *some* potential practical consequences, even if they are stylized. Are there ways to get around this hardness in practice?

---

> ### Author Rebuttal · Authors · 2025-07-31
>
> We thank the reviewer for their comments and suggestions. The response is divided into two parts: one common section addressing general concerns, followed by answers to specific questions.
>
> ***
>
> We identified the following two points as the main concerns among the reviewers: 1) relevance to the NeurIPS community and 2) the lack of practical consequences of an hardness result. In the following, we address these two points.
>
> ## Relevance
>
> First, we emphasize that convergence‐to‐equilibrium results in multi‐agent learning occupy a central place in the NeurIPS community and in the broader ML community. Our work delivers a fundamentally negative insight: it rules out the existence of efficient algorithms for computing constrained correlated equilibria, in contrast to wide array of algorithms available in the unconstrained case. In particular, there are many recent works on no-regret algorithms for correlated equilibria, and one could expect the same kind of tools to be applicable in the constrained setting. In contrast, we show that efforts to establish positive convergence guarantees of the form “no-regret algorithms converge to constrained correlated equilibria” are going to fail. We believe it is important to broadcast this impossibility result: by doing so, we can steer the community away from unfruitful directions and toward questions where real progress can be made. The NeurIPS community is precisely where researchers most invested in proving positive convergence results gather, making our impossibility result both timely and highly relevant to their efforts.
>
> We also highlight that our results answer a known open problem within the ML community (see [1,2]). Prior work explicitly left open the complexity of finding these equilibria. We settle this question definitively in the negative. This hardness result fills a clear gap in the theory: while numerous works have established positive convergence rates for regret minimization in unconstrained settings, the constrained case remained largely open until now.
>
> Finally, it is not new for leading ML venues to publish papers whose main contribution is a hardness result, rather than a positive algorithmic guarantee. This is particularly the case in areas close to game theory. As an example, we mention the recent paper “The Complexity of Two-Team Polymatrix Games with Independent Adversaries” by Hollender et al. [3] that was very well received at ICLR 25.
>
> ## Practical relevance
>
> While focusing on a general and abstract framework, our model captures important applications in the Internet economy, such as automated bidding systems that power modern digital advertising. Today’s platforms typically deploy autobidders that act on behalf of advertisers during the real-time bidding process. Each autobidder dynamically adjusts bids and usually has to satisfy constraints specified by the advertiser, such as return‐on‐investment, budget, and cost‑per‑acquisition constraints. These autobidding systems have attracted extensive attention in machine learning, originating a rich body of both theoretical and empirical works (see, e.g., [4,5,6,7]).
>
> One key aspect of these systems is that all autobidders are deployed and run directly by the advertising platform. Therefore, platforms are usually interested in designing autobidding algorithms that converge to equilibrium over time, as this makes the system more stable, predictable, and compatible with the individual incentives of each advertiser. Our impossibility result demonstrates that, in general, convergence cannot be guaranteed. Consequently, to design autobidding platforms with provable convergence guarantees, advertising systems must employ new techniques that exploit the specific structure of their problem, since no general-purpose algorithm will suffice.
>
> Due to space constraints, this point was only briefly mentioned in the current version. We will elaborate on it in the final version of the paper.
>
>
> ## References
>
> [1] Zhang, B. H., Anagnostides, I., Tewolde, E., Berker, R. E., Farina, G., Conitzer, V., & Sandholm, T. (2025). Expected variational inequalities. ICML. 2025.
>
> [2] Bernasconi, M., Castiglioni, M., Marchesi, A., Trovo, F., & Gatti, N. (2023, July). Constrained phi-equilibria. In International Conference on Machine Learning (pp. 2184-2205). PMLR.
>
> [3] Hollender, Alexandros, Gilbert Maystre, and Sai Ganesh Nagarajan. "The Complexity of Two-Team Polymatrix Games with Independent Adversaries." ICLR. 2025.
>
> [4] Balseiro, Santiago R., and Yonatan Gur. "Learning in Repeated Auctions with Budgets: Regret Minimization and Equilibrium." Proceedings of the 2017 ACM Conference on Economics and Computation. 2017.
>
> [5] Paes Leme, R., Piliouras, G., Schneider, J., Spendlove, K., & Zuo, S. (2024, July). Complex dynamics in autobidding systems. In Proceedings of the 25th ACM Conference on Economics and Computation (pp. 75-100).
>
> [6] Balseiro, S., Deng, Y., Mao, J., Mirrokni, V., & Zuo, S. (2021). Robust auction design in the auto-bidding world. Advances in Neural Information Processing Systems, 34, 17777-17788.
>
> [7] Agarwal, D., Ghosh, S., Wei, K., & You, S. (2014, August). Budget pacing for targeted online advertisements at linkedin. In Proceedings of the 20th ACM SIGKDD international conference on Knowledge discovery and data mining (pp. 1613-1619).
>
> ***
>
> ## Specific answers
>
> We thank the reviewer for the positive evaluation of our contributions.
>
> > What are the practical consequences of this work?
>
> For the practical consequences of our result, we refer to the common answer.
>
> > Are there other relaxations of constrained equilibria that may be easy to compute?
>
> A natural, but significantly weaker, relaxation of our solution concept would be to consider a randomized version of our equilibria, making the set of equilibria a convex set. This seems to align with a recent line of research: [Zhang, Brian Hu, et al. "Expected Variational Inequalities." ICML 2025.]

---

> > ### Comment · Reviewer_QrtF · 2025-08-07
> >
> > Thank you for responding to my questions -- I am satisfied.

---

### Decision · Program_Chairs · 2025-09-17

**Decision:**

Accept (poster)

**Comment:**

This paper studies the computational complexity of computing Correlated constrained equilibria in generalized games, showing hardness results.

There is general agreement among reviewers that the result is interesting, novel. There are some issues on practical relevance of the results, since the paper is mainly theoretical. Still, this does not undermine the strengths of the paper. Anyway, I would require the authors to explain this point of relevance to ML very clearly – that their results are relevant to learning dynamics, which is a fundamental thing, and explain what precisely is being ruled out (only computationally efficient dynamics, and what type(s) of convergence).